# VIKI-R: Coordinating Embodied Multi-Agent Cooperation via Reinforcement Learning

**Li Kang**[1,2*], **Xiufeng Song**[1,2*], **Heng Zhou**[3,2*], **Yiran Qin**[2,5†],
**Jie Yang**[5], **Xiaohong Liu**[1], **Philip Torr**[4], **Lei Bai**[2†], **Zhenfei Yin**[4†]

[1]Shanghai Jiao Tong University [2]Shanghai Artificial Intelligence Laboratory
[3]University of Science and Technology of China [4]University of Oxford
[5]The Chinese University of Hong Kong, Shenzhen
{faceong02, sparklexfantasy, hengzzzhou}@gmail.com
* Equal contribution [†] Corresponding author
https://faceong.github.io/VIKI-R/

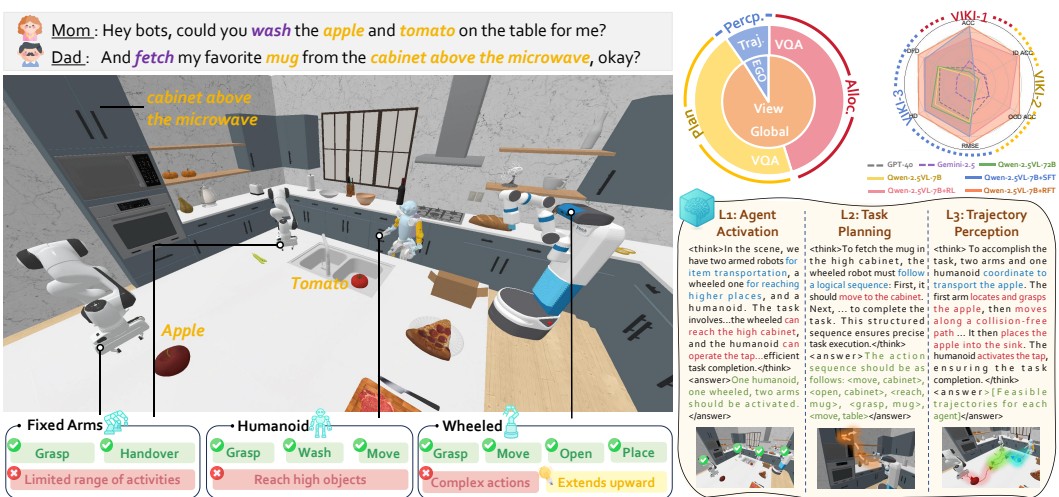

Figure 1: Embodied multi-agent cooperation involves two key aspects: (1) cross-embodiment collaboration, where different embodiments are required for different tasks (e.g., washing requires a humanoid, while only wheeled robots can fetch from high cabinets); and (2) efficient coordination, where agents work in parallel (e.g., multiple arms passing apples while a humanoid washes them) to improve overall efficiency. To support such fine-grained teamwork, we propose *VIKI-Bench*, which structures the process into three levels of visual reasoning: Level 1 – agent activation, Level 2 – task planning, and Level 3 – trajectory perception, aiming to realize an embodied multi-agent system.

## Abstract

Coordinating multiple embodied agents in dynamic environments remains a core challenge in artificial intelligence, requiring both perception-driven reasoning and scalable cooperation strategies. While recent works have leveraged large language models (LLMs) for multi-agent planning, a few have begun to explore vision-language models (VLMs) for visual reasoning. However, these VLM-based approaches remain limited in their support for diverse embodiment types. In this work, we introduce *VIKI-Bench*, the first hierarchical benchmark tailored for embodied multi-agent cooperation, featuring three structured levels: agent activation, task planning, and trajectory perception. *VIKI-Bench* includes diverse robot

embodiments, multi-view visual observations, and structured supervision signals to evaluate reasoning grounded in visual inputs. To demonstrate the utility of *VIKI-Bench*, we propose *VIKI-R*, a two-stage framework that fine-tunes a pretrained vision-language model (VLM) using Chain-of-Thought annotated demonstrations, followed by reinforcement learning under multi-level reward signals. Our extensive experiments show that *VIKI-R* significantly outperforms baselines method across all task levels. Furthermore, we show that reinforcement learning enables the emergence of compositional cooperation patterns among heterogeneous agents. Together, *VIKI-Bench* and *VIKI-R* offer a unified testbed and method for advancing multi-agent, visual-driven cooperation in embodied AI systems.

# 1   Introduction

In the science-fiction film *I, Robot* [50], the super-computer VIKI orchestrates thousands of NS-5 robots, illustrating the extraordinary coordination capabilities of heterogeneous robotic agents. This fictional depiction highlights a fundamental challenge in artificial intelligence: enabling multiple embodied agents to collaborate in dynamic, real-world environments. As illustrated in Fig. 1, addressing this challenge is critical for advancing multi-agent systems capable of achieving effective, large-scale coordination: (1) Real-world tasks often necessitate specialized embodiments—for instance, reaching high cabinets may call for a robot with extended reach, while delicate tasks demand manipulators with fine-grained control. (2) Cooperative behaviors substantially enhance task efficiency through parallelization and mutual assistance.

Recent advances have demonstrated the potential of large language models (LLMs) in enabling multi-agent planning [6, 8, 59, 37]. While these LLM-based approaches have made significant progress in high-level coordination, only a few works have explored the use of vision-language models (VLMs) for perception-driven reasoning [27, 48, 60]. However, existing VLM-based methods remain limited by the lack of embodiment diversity. As a result, the ability to reason about visual observations in heterogeneous multi-agent settings remains an underexplored challenge.

To address these gaps, we introduce **VIKI-Bench**, a comprehensive benchmark for evaluating collaborative capabilities in embodied multi-agent systems. As illustrated in Fig. 1, *VIKI-Bench* is designed around three levels of task: Agent Activation, Task Planning, and Trajectory Perception. Each task provides multi-view visual input and incorporates a diverse set of heterogeneous robots. Moreover, *VIKI-Bench* provides a multi-dimensional evaluation framework that assesses execution feasibility, task completion and planning efficiency. To the best of our knowledge, *VIKI-Bench* is the first comprehensive benchmark specifically designed to evaluate the reasoning capabilities of VLMs in hierarchical embodied multi-agent cooperation.

To advance reasoning capabilities in the multi-agent system, we introduce **VIKI-R**, a VLM-based framework that fosters reasoning abilities in multi-agent cooperation. Inspired by [13, 28, 44], our approach first grounds a pretrained VLM in task understanding through Chain-of-Thought annotations, then optimizes it via Reinforcement Learning, leveraging hierachical supervision in *VIKI-Bench*. Extensive experimental results demonstrate that *VIKI-R* significantly outperforms baseline methods across all three task levels, highlighting the effectiveness of the proposed approach.

In summary, the main contributions of this paper are as follows:

⬦ We introduce *VIKI-Bench*, the first hierarchical benchmark for embodied multi-agent cooperation, which consists of three structured task levels: agent activation, high-level task planning, and low-level trajectory perception. The benchmark features heterogeneous robot types, multi-view visual inputs, and structured supervision signals to enable comprehensive evaluation.

⬦ We propose *VIKI-R*, a two-stage learning framework that enhances visual reasoning capabilities in embodied multi-agent systems by using hierarchical reward signals to learn structured reasoning across diverse tasks, enabling generalizable cooperation in complex environments.

⬦ Extensive experimental results demonstrates the effectiveness of *VIKI-R* in *VIKI-Bench*. Our analysis highlights the importance of hierarchical supervision and reveals how reinforcement learning facilitates the emergence of compositional collaboration patterns in embodied environments.

Table 1: **Comparison to similar embodied benchmarks.** We compare VIKI-Bench to embodied AI benchmarks, focusing on natural language and multi-agent collaboration tasks. [Keys: **Views:** EGO (Ego-centric view), GL (Global view). **H.E.:** Coordination among Heterogeneous Embodiments. ]

|  | Environment | Language | Visual | Views | H.E. | Tasks Num |
|---|---|---|---|---|---|---|
| Overcooked [7] | 2D | ✓ |  | - |  | 4 |
| RoCo [31] | 3D | ✓ |  | - |  | 6 |
| WAH [36] | 3D | ✓ |  | - |  | 1,211 |
| Co-ELA [59] | 3D | ✓ |  | - |  | 44 |
| FurnMove [19] | 3D | ✓ | ✓ | EGO |  | 30 |
| PARTNR [8] | 3D | ✓ |  | - | ✓ | 100,000 |
| RoboCasa [32] | 3D | ✓ | ✓ | EGO | ✓ | 100 |
| LLaMAR (MAP-THOR) [33] | 3D | ✓ | ✓ | EGO, GL | ✓ | 225 |
| **VIKI-Bench (Ours)** | 3D | ✓ | ✓ | EGO, GL | ✓ | 23,737 |

## 2 Related Work

**Embodied Multi-Agent Cooperation**    Real-world embodied tasks often require cooperation among multiple agents. Existing studies [2, 14, 24, 30, 38, 40, 58, 67, 65, 68, 69] have explored this problem in various application domains. Research focuses on multi-agent task allocation [25, 34, 47] and joint decision-making [46, 59]. A significant body of recent work [6, 15, 22, 32, 41, 63, 70] leverages large language models (LLMs) to handle high-level reasoning and planning. Some recent works leverage video generation models [5, 39, 53, 54, 55, 56] to construct multi-agent world models [60], achieving promising results on specific tasks. However, these approaches lack visual grounding, limiting their ability to reason about spatial constraints and perceptual affordances. While a few recent efforts [48, 64, 66] incorporate vision-language models (VLMs) to obtain a more grounded understanding of the environment, research on heterogeneous multi-agent cooperation remains sparse—particularly in settings requiring fine-grained visual reasoning and embodied perception. In contrast, our work incorporates both agent heterogeneity and visual reasoning to support complex, perception-driven collaboration.

**Visual Reasoning**    Visual reasoning requires vision-language models (VLMs) to interpret and reason over visual observations to perform complex tasks. It has been applied in areas such as geometric problem-solving [12, 43, 61], robotic [16, 17, 20, 66] and scientific research [21, 29, 62]. Previous work has explored enhancing visual reasoning in VLMs through multi-stage supervision. For example, LLaVA-CoT [51] applies multi-stage supervised fine-tuning (SFT) with chain-of-thought [49] prompting. With the introduction of a rule-based reinforcement learning (RL) method, DeepSeek-R1 [13] demonstrates significant improvements in reasoning performance. Recent works [26, 28, 44] incorporate RL to further enhance visual reasoning capabilities. Our work shows that R1-style methods perform better in multi-agent embodied visual reasoning tasks.

**Embodied multi-agent benchmarks**    Recent research [1, 7, 59, 8, 32] has developed several embodied multi-agent benchmarks to evaluate collaborative behaviors. In 2D environments, LLM-Co [1] and Overcooked [7] study coordination in game play, but the simplified 2D settings limit their abilities in physical interaction. For 3D environments, a thread of work has focused on language-guided cooperative planning for embodied tasks. For instance, WAH [36] examines social intelligence in household scenarios. PARTNR [8] evaluates visual planning and reasoning under LLM-based evaluation. Other benchmarks target multi-agent manipulation. RocoBench [31] conducts object interaction tasks within a tabletop environment. FurnMove [19] requires collaboration on synchronized furniture arrangement. LLaMAR [33] focuses on multi-agent task planning and refinement, with 225 task instances, emphasizing task planning and verification. VIKI-Bench goes a step further by offering a broader evaluation framework, featuring over 23,000 task instances across 100 diverse scenes and multiple robot types that bridges both planning and manipulation domains.

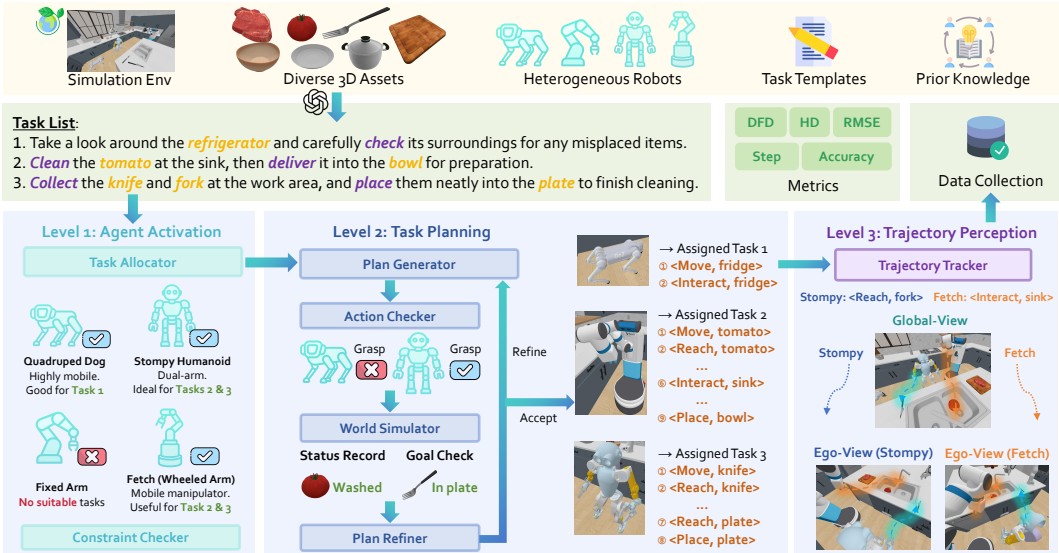

Figure 2: Overview of *VIKI-Bench*. *VIKI-Bench* is a hierarchical benchmark for evaluation on multi-agent embodied cooperation, featuring visual reasoning tasks in three levels: (1) Agent Activation, where robots are selected based on the scene image and the task context; (2) Task Planning, where a structured multi-agent action plan is generated, verified, and refined; and (3) Trajectory Perception, where the fine-grained motion trajectory of each agent is tracked from egocentric views. The benchmark involves diverse robot types and complex 3D environments, with multiple metrics for quantitative evaluation.

## 3 VIKI-Bench

### 3.1 Overview

We introduce **VIKI-Bench**, a hierarchical benchmark for studying visual reasoning in embodied multi-agent collaboration, as illustrated in Fig. 2. *VIKI-Bench* covers three levels of tasks: (1) Agent Activation, which selects appropriate agents to activate by considering the task description and the scene image; (2) Task Planning, which requires generating an ordered sequence of action primitives of multiple agents; and (3) Trajectory Perception, which involves predicting the motion trajectories of all agents. Each task includes a language instruction, with global visual observations provided for the first two levels, and egocentric views used for the trajectory perception level. Spanning thousands of tasks across heterogeneous robot morphologies and diverse household-to-industrial layouts, *VIKI-Bench* offers a concise yet comprehensive benchmark for scalable multi-agent cooperation.

### 3.2 Data Generation

#### 3.2.1 Agent Activation

We formulate the agent activation task as a visual reasoning problem, where the task allocator selects a set of appropriate robots among all agents to complete the task. Each sample is formatted as an instruction-question pair, consisting of an image observation $O$ and a task instruction $I$. The expected answer is a set of selected agents $R = \{r_j\}, j \in [1, M]$ chosen from the visible agent pool $\mathcal{A}_{\text{visible}}$ based on embodiment reasoning and task affordance.

To generate ground truth labels, we construct task-specific templates that specify which agent types are required or not required for solving the task, given the task goal and environmental context. These templates are grounded in embodiment rules and capability-based constraints (*e.g.*, mobile agents for navigation, dual-arm agents for bimanual manipulation).

To encourage interpretable reasoning, we adopt a chain-of-thought format in which the model is expected to: (1) analyze the task requirements, (2) visually identify the robots present, (3) assess each robot's suitability, and (4) conclude the final selection. For data generation, we employ GPT-4o [35]

as the task allocator $g_{\text{act}}$, prompting it with the task template and the corresponding image context. The activation result is then obtained as $R = g_{\text{act}}(I, O)$. A verification module $C_{\text{act}}$ is used to automatically check whether the generated labels conform to embodiment-grounded task constraints, followed by human inspection to correct failure cases and ensure overall label quality.

### 3.2.2 Task Planning

We construct task planning data as question-answer pairs according to the environment and specific instructions. To describe high-level operations of agents in the environment, we design basic primitive set $P$ (*e.g.*, move, grasp, etc.) as the atomic operations of all agents. The planning answer is designed as a sequence of action descriptions $A = \{a_1, a_2, ...a_N\}$, where $N$ is the length of the sequence. Each action description is formed as $a_i = (r_i, t_i, p_i, d_i)$, where $r_i, t_i, p_i, d_i$ denotes the agent, the timestep, the primitive and the destination of action $a_i$, respectively.

To generate effective planning in versatile environments, we use GPT-4o as the plan generator $g_{plan}$ and introduce an iterative refinement process. Given an instruction $I$, the corresponding observation $O$, and the primitive set $P$, the generator first decomposes the instruction into a set of goals $G$, and generates a possible planning result $A_0$, as $A_0 = g_{plan}(I, O, P)$. Then, an Action Checker $C$ verifies the feasibility of each action based on the rules of primitives, followed by a World Simulator $S$ recording the position and status of interactive entities in the environment. Subsequently, a Plan Refiner $R$ checks the completion of the goals. For any failure in planning, the refiner provides detailed feedback as an additional instruction, which is concatenated with the original instruction for the generator to revise the planning result until success. This procedure is formulated as follows.

---

**Algorithm 1** Iterative Refinement Process

---

**Require:** Plan Generator $g_{plan}$, Instruction $I_0$, Goals $G$, Observation $O$, Primitives $P$
**Ensure:** Successful Planning $A$
1:   $Success \leftarrow False$
2:   $I \leftarrow I_0$
3: **while** $\neg Success$ **do**
4:      $A \leftarrow g_{plan}(I, O, P)$
5:      $Act\_success \leftarrow C(act), \forall act \in A$                 ▷ Action feasibility check
6:      $Status \leftarrow S(A)$
7:      $Goal\_success \leftarrow is\_successful(Status, goal), \forall goal \in G$      ▷ Goal check
8:      $Success \leftarrow Act\_success \wedge Goal\_success$
9:      $I \leftarrow I + R(Act\_success, Goal\_success)$         ▷ Update feedback instruction
10: **end while**

---

### 3.2.3 Trajectory Perception

We formulate trajectory perception in multi-agent environments as a spatial keypoint prediction problem, where the model predicts motion trajectories from egocentric observations based on the task instruction. Unlike prior work [8, 20] that focuses solely on the observing agent, our setting requires predicting both the trajectory of the ego agent and those of other visible agents to facilitate collaboration, which are referred as the *ego-trajectory* and *partner-trajectories*, respectively. Given an egocentric RGB image $I$ and an action description $a_i = (r_i, t_i, p_i, d_i)$ indicating the ongoing execution, the model predicts a set of 2D trajectories $\mathcal{L} = \{T_k\}, k \in [1, M]$, where $M$ is the number of agents in the scene, and $L_k = \{(x_j, y_j)\}_{j=1}^{L}$ denotes a temporally ordered spatial motion for agent $r_k$ in coordinate sequences.

To construct these samples, we sample diverse egocentric observations from simulated multi-agent scenes with the corresponding task descriptions. Based on the egocentric observations and detailed instructions for each visible agent, the trajectory of each agent is manually annotated by formulating feasible a motion path in the form of coordinate sequences. All data undergoes human verification to ensure temporal consistency and spatial alignment with the instruction and environment.

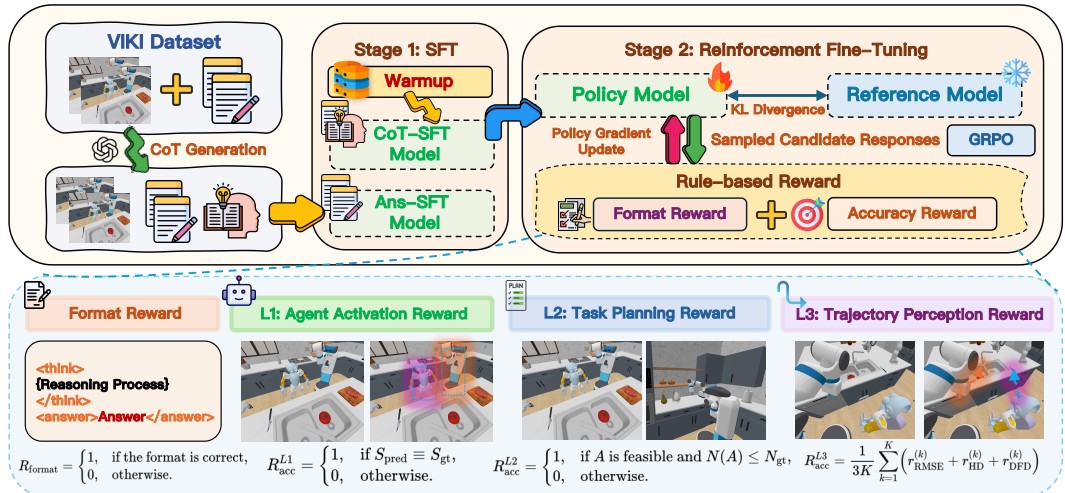

Figure 3: Framework of *VIKI-R*. We adopted supervised fine-tuning (SFT) and reinforcement fine-tuning on the VIKI dataset, incorporating format and accuracy rewards to optimize the policy model.

## 3.3 Data Statistics

The VIKI benchmark comprises over 20,000 multi-agent task samples across 100 diverse scenes derived from the RoboCasa [32] based on ManiSkill3 [45], each with fine-grained object configurations and varied spatial layouts. The dataset involves 6 types of heterogeneous embodied agents (e.g., humanoids, wheeled arms, quadrupeds) interacting with over 1,000 unique asset combinations. Each scene provides both global and egocentric camera views to support perception and planning. More details are provided in Appendix C.1.

## 4 VIKI-R

### 4.1 Overview

We introduce *VIKI-R*, a two-stage fine-tuning framework that endows vision–language models with robust visual reasoning abilities, as shown in Fig. 3. In the first stage, *SFT-based Warmup*, the model undergoes supervised fine-tuning on high-quality Chain-of-Thought (CoT) annotations, optimizing the likelihood of both intermediate reasoning steps and final answers. This stage instructs the model to acquire domain-specific reasoning patterns. In the second stage, *Reinforcement Fine-Tuning*, the policy is refined using the Grouped Relative Proximal Optimization (GRPO) algorithm [42]. For each visual–question pair, grouped candidate answers are sampled and evaluated using a composite reward function based on answer format and correctness. Standardized advantages are then computed to guide policy updates under a KL-divergence constraint, ensuring stable and consistent improvement.

### 4.2 Training Objectives

**SFT-based Warmup**  In the first phase, we employ Supervised Fine-Tuning (SFT) with data annotated with Chain-of-Thought (CoT) reasoning process. Each training instance is denoted as $(x, q, r, a)$, where $x$ represents the visual input, $q$ the associated task, $r$ the intermediate reasoning steps, and $a$ the final answer. The SFT objective maximizes the joint likelihood of the reasoning and answer tokens conditioned on the input:

$$\mathcal{L}_{\text{SFT}} = -\mathbb{E}_{(x,q,r,a)\sim\mathcal{D}} \sum_{t=1}^{T} \log \pi_\theta\big(y_t \mid x, q, y_{<t}\big), \qquad (1)$$

where $\mathcal{D}$ is the CoT-annotated dataset, $y = [r, a]$ is the concatenated sequence of reasoning and answer tokens, and $\pi_\theta$ denotes the model's token distribution.

**Reinforcement Fine-Tuning**  Starting from the Chain-of-Thought initialized policy $\pi_{\text{CoT}}$, we perform group-relative reinforcement fine-tuning following the GRPO formulation. Given an input $s = (x, q)$, we sample a group of $G$ candidate outputs $\{a_i\}_{i=1}^{G}$, each receiving a reward $R_i$. We compute the empirical mean $\bar{R}$ and standard deviation $\sigma_R$ of the group, and define the standardized advantage for each candidate as

$$\mathcal{A}_i = \frac{R_i - \bar{R}}{\sigma_R}. \tag{2}$$

This group-relative normalization highlights candidates that outperform their peers and stabilizes learning by removing dependence on absolute reward scale.The probability ratio between the current policy $\pi_\theta$ and the reference policy $\pi_0$ (initialized from $\pi_{\text{CoT}}$) is defined as

$$r_i(\theta) = \frac{\pi_\theta(a_i \mid s)}{\pi_0(a_i \mid s)}. \tag{3}$$

The clipped surrogate objective is then given by

$$\mathcal{L}^{\text{CLIP}}(\theta) = \mathbb{E}_s \left[ \sum_{i=1}^{G} \min \left( r_i(\theta)\mathcal{A}_i, \ \text{clip}\left(r_i(\theta), 1 - \epsilon, 1 + \epsilon\right)\mathcal{A}_i \right) \right], \tag{4}$$

where $\epsilon$ is the clipping coefficient that bounds the policy update step.

Finally, the policy is optimized under a KL-divergence constraint to maintain proximity to the reference policy:

$$\mathcal{J}(\theta) = \mathcal{L}^{\text{CLIP}}(\theta) - \beta \, \text{D}_{\text{KL}}\big(\pi_\theta(\cdot \mid s) \,\|\, \pi_0(\cdot \mid s)\big), \tag{5}$$

where $\beta > 0$ controls the trust region enforced by the KL regularizer. This GRPO formulation enables stable and sample-efficient reinforcement fine-tuning, allowing the policy to leverage group-relative advantages for compositional spatial reasoning.

## 4.3  Reward Design

To guide the model towards both structured output and task accuracy, we formulate the overall reward into a *format reward* and a task-specific *accuracy reward*, as:

$$R = \lambda_1 \times R_{\text{format}} + \lambda_2 \times R_{\text{acc}}, \tag{6}$$

where $R_{\text{format}}$ enforces the output format and $R_{\text{acc}}$ corresponds to the three subtask rewards, as defined below. $\lambda_1$ and $\lambda_2$ refer to the weights of both rewards, respectively.

**Format Reward**  To encourage explicit reasoning, we assign a binary format reward: the model receives 1 point if it correctly encloses the intermediate reasoning steps within `<think>...</think>` and the final answer within `<answer>...</answer>`, and 0 otherwise. By enforcing these tags, we prompt the model to articulate its chain-of-thought before delivering the answer, thereby improving interpretability and guiding systematic reasoning.

**Agent Activation Reward**  We define the agent activation reward as an exact-match indicator between the predicted agent set $S_{\text{pred}}$ and the ground-truth set $S_{\text{gt}}$:

$$R_{\text{acc}}^{L1} = \begin{cases} 1, & \text{if } S_{\text{pred}} \equiv S_{\text{gt}}, \\ 0, & \text{otherwise.} \end{cases} \tag{7}$$

**Task Planning Reward**  While multiple feasible plans may exist, we define the reward to favor efficient solutions. Specifically, a predicted plan $A$ only receives the reward if it is feasible and its length does not exceed that of the ground-truth plan $N_{\text{gt}}$. Let $N(A)$ denote the length of the predicted action sequence $A$, the task planning reward is defined as:

$$R_{\text{acc}}^{L2} = \begin{cases} 1, & \text{if } A \text{ is feasible and } N(A) \leq N_{\text{gt}}, \\ 0, & \text{otherwise.} \end{cases} \tag{8}$$

Details on how plan feasibility is checked are provided in Appendix C.2.

**Trajectory Perception Reward** Let $P^{(k)} = \{p_t^{(k)}\}_{t=1}^T$ and $G^{(k)} = \{g_t^{(k)}\}_{t=1}^T$ denote the predicted and ground-truth trajectories for agent $k$, respectively. To evaluate trajectory prediction quality for each agent $k$, we compute three normalized standard geometric distance metrics between the predicted trajectory and the ground-truth trajectory: Root Mean Square Error (RMSE, denoted as $\hat{d}_{\text{RMSE}}$), Hausdorff Distance (HD, denoted as $\hat{d}_{\text{HD}}$)[18], and Discrete Fréchet Distance (DFD, denoted as $\hat{d}_{\text{DFD}}$)[11]. Since smaller distances indicate better alignment between predicted and ground-truth trajectories, we transform the distance $\hat{d}$ into a reward-like score using the transformation $r = 1 - \hat{d}$. The final trajectory perception reward is defined as:

$$R_{\text{acc}}^{L3} = \frac{1}{3K} \sum_{k=1}^{K} \left( r_{\text{RMSE}}^{(k)} + r_{\text{HD}}^{(k)} + r_{\text{DFD}}^{(k)} \right),\tag{9}$$

## 5 Experiments

Table 2: Performance comparison across the three hierarchical task levels of VIKI-Bench. Best scores are highlighted in **bold**, and the second-best scores are underlined.

| Method | VIKI-L1 | VIKI-L2 | | | VIKI-L3 | | | |
|---|---|---|---|---|---|---|---|---|
| | $\text{ACC}_{\text{ID}}\uparrow$ | $\text{ACC}_{\text{ID}}\uparrow$ | $\text{ACC}_{\text{OOD}}\uparrow$ | $\text{ACC}_{\text{AVG}}\uparrow$ | RMSE $\downarrow$ | HD $\downarrow$ | DFD $\downarrow$ | AVG $\downarrow$ |
| **Closed-Source Models** | | | | | | | | |
| GPT-4o | 18.40 | 22.56 | 10.02 | 17.50 | 100.80 | 115.34 | 131.05 | 115.73 |
| Claude-3.7-Sonnet | 12.40 | 19.44 | 0.57 | 11.82 | 283.31 | 323.53 | 346.88 | 317.91 |
| Gemini-2.5-Flash-preview | 31.40 | 20.00 | 10.51 | 16.17 | 453.89 | 519.14 | 540.80 | 504.61 |
| **Open-Source Models** | | | | | | | | |
| Qwen2.5-VL-72B-Instruct | 11.31 | 8.40 | 1.20 | 5.49 | 81.31 | 94.62 | 113.15 | 96.36 |
| Qwen2.5-VL-32B-Instruct | 9.50 | 3.60 | 0.00 | 2.15 | 88.48 | 99.80 | 119.78 | 102.69 |
| Llama-3.2-11B-Vision | 0.40 | 0.50 | 0.00 | 0.30 | 192.69 | 223.57 | 231.85 | 216.04 |
| **Qwen2.5VL-3B-Instruct** | | | | | | | | |
| Zero-Shot | 1.95 | 0.22 | 0.00 | 0.13 | 96.22 | 114.93 | 130.98 | 114.04 |
| +Ans SFT | 35.29 | 81.06 | 30.71 | 60.74 | 74.70 | 90.28 | 102.26 | 89.08 |
| +VIKI-R-Zero | 20.40 | 0.00 | 0.00 | 0.00 | 80.36 | 95.36 | 120.27 | 98.66 |
| +VIKI-R | 74.10 | 93.61 | 32.11 | 68.78 | 75.69 | 90.25 | 103.65 | 89.86 |
| **Qwen2.5VL-7B-Instruct** | | | | | | | | |
| Zero-Shot | 4.26 | 0.44 | 0.00 | 0.26 | 81.93 | 103.82 | 112.91 | 99.55 |
| +Ans SFT | 72.20 | **96.89** | 25.62 | 68.13 | 65.32 | 81.20 | 90.89 | 79.14 |
| +VIKI-R-Zero | **93.59** | 0.17 | 0.00 | 0.10 | 67.42 | 85.30 | 95.32 | 82.68 |
| +VIKI-R | 93.00 | 95.22 | **33.25** | **69.25** | **64.87** | **79.23** | **89.36** | **77.82** |

### 5.1 Experimental Setup

**Training Paradigms and Baselines** To assess the impact of different training strategies on performance and generalization, we compare the following methods: (1)Ans-SFT: a supervised fine-tuning (SFT) approach focusing solely on answer generation. (2)VIKI-R-Zero: a reinforcement learning (RL) variant that applies GRPO directly, without any prior CoT activation. (3)VIKI-R: our two-phase scheme—first SFT on a small CoT-annotated subset, followed by GRPO-based RL. All variants use Qwen2.5-VL-Instruct [4] as the base model in both 3B and 7B sizes to study the effect of model scale. For a comprehensive comparison, we include open-source models[4, 23]and leading closed-source systems GPT-4o [35], gemini-2.5-flash-preview [9]) and claude-3.7-sonnet[3] as baselines. Detailed hyperparameters and additional setup information are provided in Appendix D.

**Evaluation Metrics** We adopted task-specific metrics to evaluate performance across the three stages of the *VIKI-Bench*. For agent activation (VIKI-L1), we report classification accuracy based on whether the selected agents match the ground truth. For task planning (VIKI-L2), we evaluate accuracy based on whether the predicted plan is both feasible and no longer than the ground-truth plan, reflecting correctness and execution efficiency. For trajectory perception (VIKI-L3), we evaluate the predicted trajectories using RMSE, Hausdorff Distance (HD) [18] and Discrete Fréchet Distance (DFD) [11], which measure spatial and temporal alignment with ground-truth motion paths.

## 5.2 Overall Performance Analysis

Tab. 2 highlights three main observations. First, when comparing open-source and closed-source models under zero-shot evaluation (without any *VIKI-Bench* training), closed-source models hold a clear advantage. Among closed-source systems, Gemini-2.5-Flash-preview achieves the highest agent activation accuracy, while GPT-4o excels at trajectory perception. In contrast, both Gemini and Claude exhibit almost no trajectory-prediction capability. Second, the model scale critically affects open-source VLM performance. The 72B-parameter Qwen2.5-VL matches or even surpasses some closed-source baselines on perception metrics, but reducing the model to 32B parameters incurs substantial drops in both planning accuracy and trajectory quality. This underscores the importance of model capacity for handling complex multi-agent visual reasoning. Third, our two-stage fine-tuning framework *VIKI-R* outperforms purely supervised Ans-SFT and VIKI-R-zero. While Ans-SFT yields strong in-domain improvements, it fails to generalize to out-of-domain scenarios. These results confirm that integrating reinforcement learning substantially enhances visual reasoning capabilities in hierarchical multi-agent cooperation.

## 5.3 Feedback-Driven Iterative Refinement

We compare two planning strategies: standard sampling (up to $k$ attempts without guidance) and feedback-driven sampling (injecting feedback between attempts). Tab. 3 demonstrates the impact of feedback-driven sampling. By injecting feedback between failed attempts, GPT-4o achieves improvements of 1.9% at `pass@3` and 3.6% at `pass@6`. Claude-3-7-Sonnet sees gains of 1.5% and 2.3% and Gemini-2.5-Flash records increases of 1.8% and 3.0%. On average, feedback-driven sampling boosts `pass@3` by 1.7% and `pass@6` by 3.0%, highlighting that iterative feedback effectively steers the model away from repeated mistakes and yields more reliable plans.

Table 3: Task planning success rates (%) under two sampling strategies. `pass@k` denotes the probability of obtaining at least one valid plan within $k$ independent attempts, while `pass@k_fb` is measured when feedback is appended after each failed attempt.

| Model | pass@1 | pass@3 | pass@3_fb | pass@6 | pass@6_fb |
|---|---|---|---|---|---|
| GPT-4o [35] | 18.4 | 18.7 | 20.6 | 18.7 | 22.3 |
| Claude-3.7-Sonnet [3] | 12.4 | 12.4 | 13.9 | 12.5 | 14.8 |
| Gemini-2.5-Flash-preview [9] | 31.4 | 31.6 | 33.4 | 31.7 | 34.7 |

## 5.4 Ablation Study

We conduct extensive ablation studies to understand the design choices in VIKI-R, covering reward shaping, model scaling, prompt variants, and comparisons with relevant baselines. Detailes are provided in Appendix G. In this section, we highlight one instructive ablation on the step penalty.

Tab. 4 demonstrates the impact of introducing a constraint-based step penalty. With the step penalty enabled, VIKI-R improves accuracy by 39.7% and 88.0% on out-of-domain and in-domain tasks, indicating substantially better generalization and execution correctness. Meanwhile, the average action length decreases by 1.92 steps, showing that penalizing unnecessary steps effectively encourages concise plans. Overall, the step penalty promotes more transferable and efficient planning strategies.

Table 4: Effect of the step penalty on 1,000 challenging reasoning tasks sampled from both the out-of-domain (OOD-H) and in-domain (ID-H) splits. $\Delta$ Steps measures the average difference between the action length of predicted plan and the ground-truth plan.

| Variant | $\text{ACC}_{\text{OOD-H}} \uparrow$ | $\text{ACC}_{\text{ID-H}} \uparrow$ | $\Delta \text{Steps} \downarrow$ |
|---|---|---|---|
| VIKI-R (without step penalty) | 7.1% | 8.0% | 1.97 |
| VIKI-R (with step penalty) | **46.8%** (+39.7%) | **96.0%** (+88.0%) | **0.05** (-1.92) |

## 5.5 Real-World Case Study

To validate our approach beyond simulation, we perform real-world evaluations on **VIKI-L1** tasks using a dual-arm RealMan robot with an Agilex mobile base and two Franka Research 3 arms. We

instantiate 100 tasks with GPT-4o [35]–generated instructions and evaluate leading vision–language models in a zero-shot manner. The results can be found in the Tab. 5

Table 5: Real-world accuracy on VIKI-L1 tasks (%).

| Model | Acc. | Model | Acc. |
|---|---|---|---|
| Qwen2.5-VL-32B [4] | 8.0 | Qwen2.5-VL-72B [4] | 14.0 |
| Claude-3.7-Sonnet [3] | 15.0 | Gemini-2.5-Flash [9] | **28.0** |

These experiments confirm that while real-world execution remains challenging, current VLMs already demonstrate basic embodied reasoning and scene grounding. Bridging this gap between simulation and reality offers a promising direction for future large-scale embodied benchmarks.

### 5.6 Insights from Training

Throughout our experiments, we identified several key behaviors that illustrate both the strengths and limitations of GRPO in our hierarchical multi-agent setting.

**Dependence on Base Policy Quality**  The effectiveness of GRPO depends critically on the competence of the pretrained policy. In VIKI-L2 planning, the zero-shot model produces almost no valid plans, and VIKI-R-Zero yields negligible improvement. By contrast, in the VIKI-L1 activation and VIKI-L3 perception tasks where the base policy already generates some correct responses—GRPO delivers clear performance gains. These observations indicate that reinforcement-based fine-tuning requires an initial set of correct rollouts to guide effective policy updates.

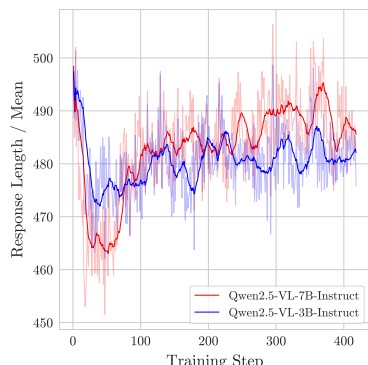

Figure 4: Response length of the Qwen2.5-VL-3B/7B-Instruct model at training time.

**Evolution of Response Length**  We tracked the average token length of model outputs during VIKI-R training in Fig. 4. In the early stages, output length decreases as the model prioritizes format compliance to secure the format reward. Once format accuracy saturates, the policy shifts focus toward maximizing task correctness, and output length gradually increases to include the necessary reasoning details.

### 5.7 Evaluation with Human Experts

We further compare VIKI-R with human experts. As shown in Tab. 6, humans consistently achieve strong performance across all VIKI levels. Meanwhile, after training, VIKI-R attains human-comparable capabilities, substantially narrowing the gap on both success rates and reasoning-consistency metrics. This suggests the practical potential of human-agent collaboration, where VIKI-R can serve as a reliable partner to assist human decision-making in embodied planning.

Table 6: Human expert and VIKI-R comparison on VIKI-Bench (150 samples for each tasks).

| Method | VIKI-L1↑ | VIKI-L2↑ | VIKI-L3 (RMSE)↓ | VIKI-L3 (HD)↓ | VIKI-L3 (DFD)↓ |
|---|---|---|---|---|---|
| VIKI-R | 93.00% | 69.25% | 64.87 | 79.23 | 89.36 |
| Human Expert | 98.00% | 96.50% | 48.84 | 58.97 | 71.30 |

## 6  Conclusion

This paper presents *VIKI-Bench*, a hierarchical benchmark for evaluating vision-language models in embodied multi-agent collaboration. We further introduce *VIKI-R*, a two-stage framework that combines supervised pretraining and reinforcement learning to solve multi-agent tasks across activation, planning, and perception levels. While our study focuses on simulated environments, extending this framework to real-world settings and dynamic agents remains promising future work.

## Acknowledgment

This work was supported by the Shanghai Municipal Science and Technology Major Project, a locally commissioned task from the Shanghai Municipal Government, and the Shanghai Artificial Intelligence Laboratory.

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

# A   Limitations

Although VIKI-Bench and VIKI-R advance embodied multi-agent cooperation, several challenges remain unresolved.

First, the hierarchical task levels proposed in VIKI-Bench, while useful for structuring cooperative interactions, may not fully reflect all dynamic conditions and cooperative practice in real-world scenarios. Real environments involve unforeseen obstacles, shifting objectives, and adaptive agent behaviors that are difficult to model within a fixed hierarchical framework.

Second, while VIKI-R's hierarchical reward designs effectively enhance agent performance, their effectiveness relies on multi-level fine-tuning, which raises the needs for computational costs. A possible solution is to devise a more precious reward structure that adapt to complex environmental variations without requiring excessive tuning.

Additionally, although the emergent collaborative behaviors show prominent performance in multi-level tasks, the reasoning process and interpretability remains underexplored. Such studies are crucial for ensuring trust and facilitating human-AI collaboration in real-world deployments.

In conclusion, while our benchmark and framework provide a strong foundation for multi-agent cooperation, they face challenges in adaptability, computational complexity and interpretability. Future work could focus on expanding task diversity, improving framework design, and enhancing model transparency. Addressing these limitations will be essential for developing more robust and scalable multi-agent systems.

# B   Broader Impacts

The development of VIKI-Bench and VIKI-R has significant influence on both embodied multi-agent research and real-world applications. By introducing a hierarchical benchmark for embodied multi-agent cooperation, this work enables multi-dimensional evaluation and comparison of vision-language models (VLMs) in complex, dynamic environments. In practice, VIKI-Bench can accelerate progress in real-world robotics applications, such as warehouse automation, autonomous driving, and collaborative industrial robots, where heterogeneous agents must coordinate under visual uncertainty.

The proposed VIKI-R framework demonstrates how fine-tuning VLMs with reasoning and reinforcement learning can improve multi-agent decision-making, potentially leading to more adaptable and efficient autonomous systems. However, the deployment of such systems raises important considerations, including safety, fairness in decision-making, and the potential relationships between human and robots. Future work should address these challenges while leveraging VIKI-Bench's structured supervision to ensure robustness and interpretability in real-world scenarios. Ultimately, this research paves the way for more sophisticated AI systems capable of seamless cooperation in visually rich, dynamic environments.

# C   Details of VIKI-Bench

This section introduces details of VIKI-Bench, including the overview of data statistics, plan feasibility checking and experimental setup.

## C.1   Data Statistics

The datasets in *VIKI-Bench* are organized into three hierarchical levels:

- VIKI-L1: Agent activation (10,714 samples; 8,171 training, 2,043 testing).
- VIKI-L2: Symbolic planning (10,714 samples; 7,196 in-domain training, 1,800 in-domain testing, and 1,218 out-of-domain testing from held-out scenes).
- VIKI-L3: Trajectory perception (2,309 samples; 1,767 training, 442 testing).

This multi-stage structure facilitates comprehensive evaluation of high-level coordination and low-level motion prediction in realistic, dynamic environments.

### C.2 Plan Feasibility Check

We adopt the same checking pipeline as used for ground-truth (GT) data generation (Section 3.2.2) to verify the correctness of a given plan. Given a set of task goals $G$ and a candidate plan $A$, an Action Checker $C$ first validates the feasibility of each action in $A$ according to the primitive rules and preconditions; a World Simulator $S$ then rolls out the execution of $A$ while tracking the states of interactive entities in the environment, after which we check whether every goal in $G$ is satisfied. If the plan is feasible and all goals are achieved, we return the plan length $N(A)$ as a measure of planning efficiency. Notably, our reward function includes a step penalty that requires the predicted plan to be no longer than the GT plan to receive any score, i.e., $N(A) \leq N(A_{\text{GT}})$, preventing reward hacking where a model could inflate reward by producing unnecessarily long and inefficient plans; §5.4 further substantiates the effectiveness of this design choice.

## D  Implementation Details

As summarized in Table 13, during GRPO we operate under a PPO framework using the VLLM engine with the XFORMERS attention backend. Inputs are truncated beyond a 4096-token prompt length and a 2048-token response length. We employ a total batch size of 256, subdivided into PPO minibatches of 128 and micro-batches of 10 per GPU; the actor network is optimized with a learning rate of $1 \times 10^{-6}$. KL-divergence regularization (coefficient 0.01, `low_var_kl` variant) is used, while entropy regularization and KL-based rewards are disabled. Gradient checkpointing reduces memory footprint, targeting 60% memory utilization during rollout with five rollouts per prompt, and both chunked prefill and eager execution are disabled. Finally, we train VIKI-L1 for five epochs, VIKI-L2 for fifteen epochs, and VIKI-L3 for two epochs.

We build upon the `Qwen2.5-VL-3B` and `Qwen2.5-VL-7B` backbones, orchestrating the entire pipeline with the open-source `verl` framework and employing `LLamaFactory` for supervised fine-tuning. All experiments run on a single node equipped with eight NVIDIA A800 GPUs. Three tasks—VIKI-L1, VIKI-L2, and VIKI-L3—with the following data splits: **VIKI-L1**: 10714 samples (500 cold-start SFT, 8171 training, 2043 testing);**VIKI-L2**: 10174 samples, of which 1218 are held out as OOD and the remaining 8956 as ID (with 500 cold-start CoT, 7196 training, 1800 ID testing);**VIKI-L3**: 2309 samples (100 cold-start CoT, 1767 training, 442 testing).

## E  Analysis of Training Stage of VIKI-R

As illustrated in Figure 5, the four panels reflect two distinct axes of variation: model capacity (3B in panels a–b vs. 7B in c–d) and initialization strategy (RL-only in VIKI-R-ZERO vs. SFT cold-start + RL in VIKI-R). Even without SFT, the 7B R-ZERO curve (panel c) begins at ∼0.1 and climbs to ∼0.9 by step 120, compared to ∼0.04 to ∼0.27 for the 3B R-ZERO (panel a), underscoring scale effects. However, both R-ZERO variants exhibit sluggish and oscillatory learning: the base model lacks sufficient task reasoning capacity to roll out coherent action sequences, resulting in unstable gradient signals and limited policy improvement without a prior SFT warm-up. Introducing a CoT cold-start further boosts performance: the 3B VIKI-R variant (panel b) launches at ∼0.3 and reaches ∼0.85 by step 140—substantially outpacing its R-ZERO counterpart—while the 7B VIKI-R (panel d) jumps in at 0.65 and exceeds ∼0.95 by step 70. Taken together, these results show that both larger capacity and SFT initialization independently accelerate learning, and when combined, yield the fastest convergence and highest final rewards.

Figure 6 juxtaposes the planning task dynamics across both model sizes and initialization strategies. On the 3B backbone, the RL-only R-ZERO variant (panel a) starts near a negligible mean reward (0.009), briefly peaks at ∼0.019 around step 50, then settles to 0.011 with persistent fluctuations—indicative of learning but limited headroom. This poor performance stems from the base model's limited reasoning capacity, which fails to produce coherent rollout trajectories without SFT initialization, yielding weak reward signals. In contrast, the CoT-initialized VIKI-R (panel b) launches at ∼0.56 (reflecting SFT warm-up) and swiftly climbs to ∼0.92 by step 60, eventually plateauing around ∼0.94 with minimal oscillation, demonstrating dramatically accelerated and stable policy improvement. For the 7B backbone, R-ZERO (panel c) begins at 0.06 and converges to ∼0.075 by step 20, maintaining a narrow band around that value thereafter. The 7B-R design (panel d), however, initiates at ∼0.45 and reaches ∼0.90 by step 60, then settles around ∼0.92, mirroring the

3B pattern of a strong SFT head start followed by rapid RL fine-tuning. These results confirm that both increased model capacity and SFT cold-start independently enhance planning performance, with their combination yielding the most pronounced gains.

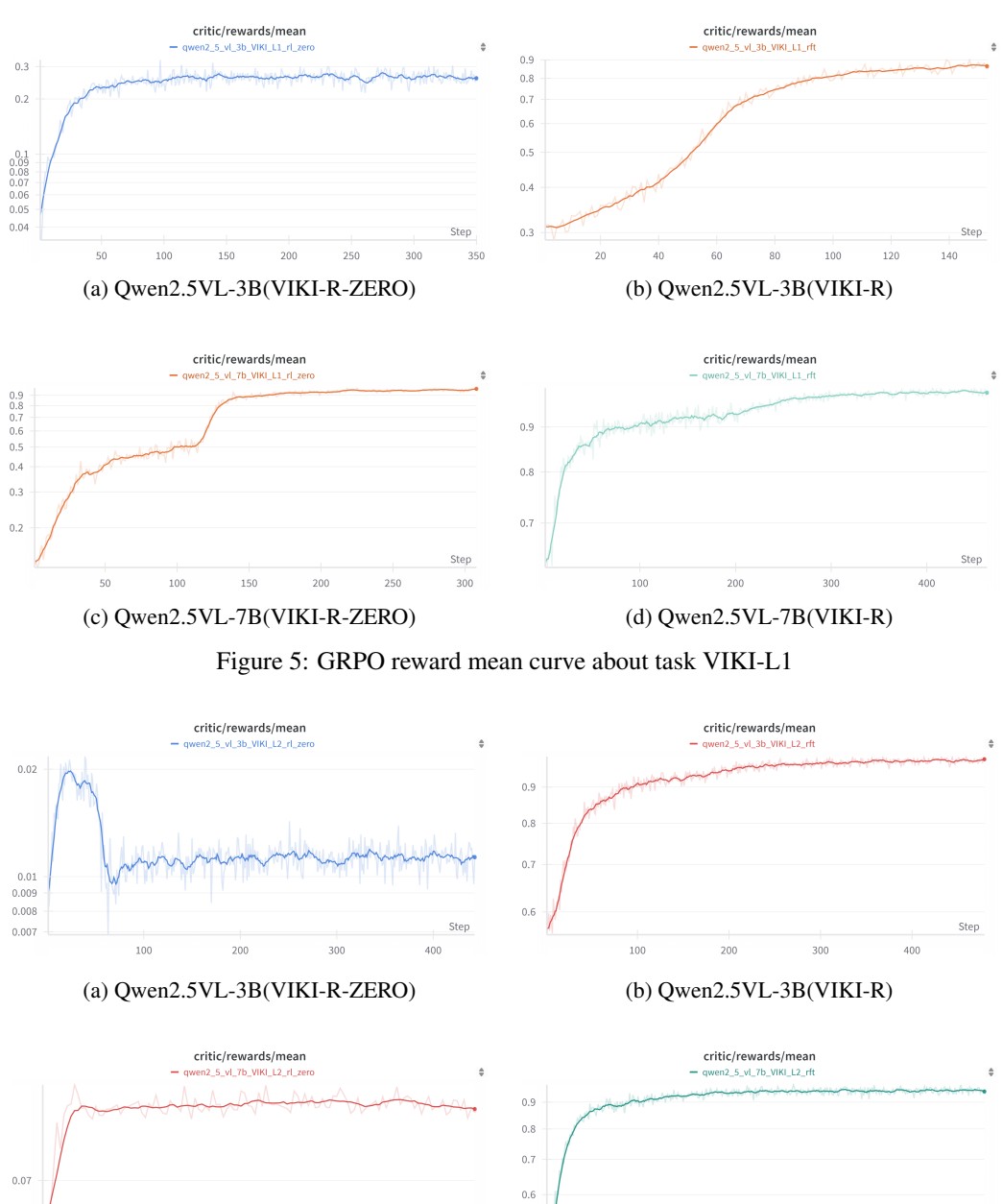

Figure 5: GRPO reward mean curve about task VIKI-L1

Figure 6: GRPO reward mean curve about task VIKI-L2

In the trajectory execution task (Fig. 7), both initialization strategies achieve high asymptotic rewards, demonstrating strong local motion capability. On the 3B backbone, R-ZERO rises from ~0.12 to 0.83, while CoT-initialized R starts higher (~0.45) and converges faster to 0.84. On the 7B model, both show similar trends, with R-ZERO reaching 0.80 and R stabilizing around 0.75 after a brief dip.

Overall, trajectory execution depends less on high-level planning—RL-only training achieves strong performance, while SFT initialization mainly accelerates early convergence.

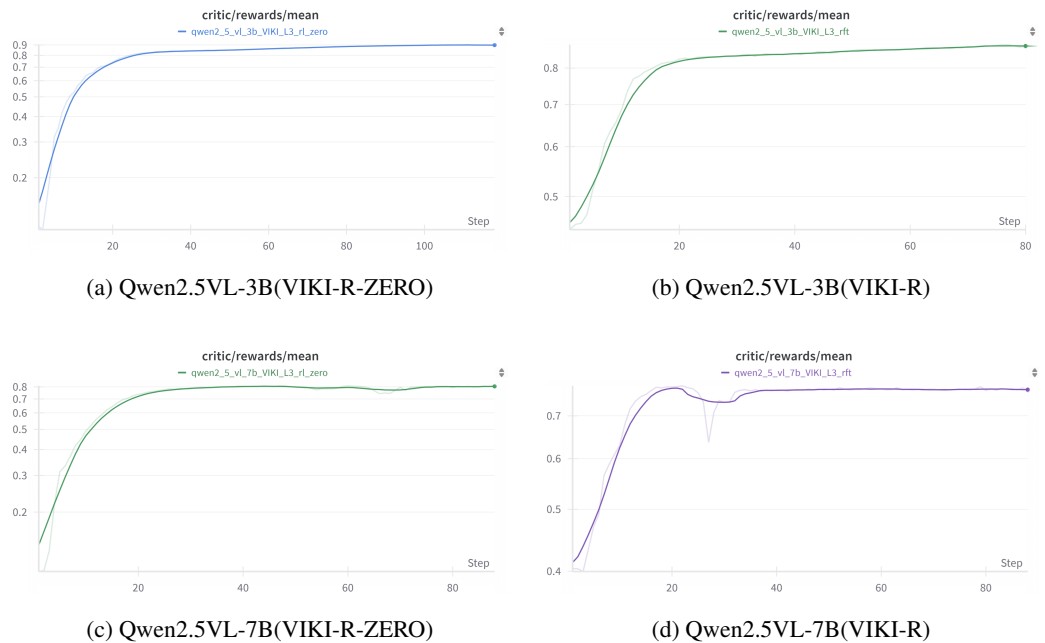

(a) Qwen2.5VL-3B(VIKI-R-ZERO)

(b) Qwen2.5VL-3B(VIKI-R)

(c) Qwen2.5VL-7B(VIKI-R-ZERO)

(d) Qwen2.5VL-7B(VIKI-R)

Figure 7: GRPO reward mean curve about task VIKI-L3

## F    Data Demonstration

Fig. 8 provides additional data visualizations showcasing qualitative demos of VIKI-L1 and VIKI-L2.

## G    Additional Ablation Studies

We further perform four complementary ablation studies to better understand the design choices in **VIKI-R**, focusing on reward structure, model scale, prompt formulation, and policy optimization.

**1. Reward Structure.**    We evaluate the effect of removing the step-level reward in the **VIKI-L2** dataset. As shown in Table 7, removing this component leads to a consistent performance drop on both in-distribution (ID) and out-of-distribution (OOD) splits, confirming that stepwise feedback is crucial for stable reasoning.

Table 7: Effect of step reward on VIKI-L2 performance.

| Model | VIKI-L2-ID (%) | VIKI-L2-OOD (%) |
|---|---|---|
| Qwen2.5-VL-7B-Inst (with step) | 95.22 | 33.25 |
| Qwen2.5-VL-7B-Inst (w/o step) | 92.06 | 29.51 |

**2. Model Scaling.**    We assess the impact of model size using 3B, 7B, and 32B backbones. As Table 8 shows, larger models consistently outperform smaller ones, highlighting that scaling strengthens compositional reasoning and improves both low- and high-level task accuracy.

**3. Prompt Design.**    We analyze the effect of removing reasoning guidance prompts such as *"You FIRST think about the reasoning process as an internal monologue and then provide the final answer."* As shown in Tab. 9, accuracy remains nearly unchanged, though convergence becomes slower. This suggests that the reinforcement signal alone enables the model to explore reasoning paths even without explicit instruction.

Table 8: Effect of model scale on reasoning performance.

| Model | VIKI-L1 (%) | VIKI-L2 (%) |
|---|---|---|
| Qwen2.5-VL-3B-Inst | 74.10 | 68.78 |
| Qwen2.5-VL-7B-Inst | 93.00 | 69.25 |
| Qwen2.5-VL-32B-Inst | **95.30** | **74.74** |

Table 9: Effect of reasoning prompt on training efficiency.

| Model | VIKI-L1 Acc. (%) |
|---|---|
| Qwen2.5-VL-3B-Inst | 74.10 |
| Qwen2.5-VL-3B-Inst (w/o guidance) | 73.47 |
| Qwen2.5-VL-7B-Inst | **93.00** |
| Qwen2.5-VL-7B-Inst (w/o guidance) | 92.95 |

**4. Optimization Algorithm.** We compare our GRPO-based optimization with the open-source **DAPO** [57] framework. Results in Tab. 10 show minimal accuracy differences, demonstrating that our compositional reinforcement learning pipeline is robust to alternative PPO-style optimizers.

Table 10: Comparison between GRPO and DAPO on VIKI-L2.

| Model | VIKI-L2-ID (%) | VIKI-L2-OOD (%) |
|---|---|---|
| Qwen2.5-VL-3B-Inst | 93.61 | 32.11 |
| Qwen2.5-VL-3B-Inst (DAPO) | 93.22 | 33.33 |
| Qwen2.5-VL-7B-Inst | 95.22 | 33.25 |
| Qwen2.5-VL-7B-Inst (DAPO) | 95.44 | 33.58 |

Overall, these studies collectively confirm that VIKI-R benefits from stepwise reward shaping and scaling, is largely invariant to prompt formulation, and remains stable across different optimization algorithms.

# H  Additional Experiments

**Extension of Evaluation to Other Relevant Methods.** To further contextualize our benchmark, we extended evaluation to include several representative approaches for vision-based multi-agent collaboration, even though none of them directly align with our hierarchical coordination setting. Specifically, we adapted ReAct [52] and the Multi-Agent Debate (MAD) framework [10], aligning their input–output protocols to match the VIKI-Bench interface.

The results show that while existing approaches can handle basic visual reasoning and short-horizon interactions, they struggle with long-term coordination and compositional task planning—core aspects emphasized in VIKI-Bench. Our proposed VIKI-R framework, combining supervised fine-tuning and reinforcement learning, effectively bridges this gap by supporting structured collaboration among multiple embodied agents.

These findings highlight the growing relevance of multi-robot cooperation and demonstrate how VIKI-Bench fills a critical gap in evaluating hierarchical, vision-based coordination across agents. We expect this benchmark to serve as a foundation for future research on scalable multi-agent reasoning.

We report the computational resources and configurations used for fine-tuning VIKI-R on Qwen2.5-VL backbones. All experiments were conducted on servers with **8 × NVIDIA A800 GPUs (80GB)**. Each task level (VIKI-L1/L2/L3) was trained separately under identical optimization settings, covering both supervised fine-tuning (SFT) and reinforcement fine-tuning (GRPO). Overall, VIKI-R can be trained end-to-end within one day on a single 8-GPU node, ensuring reproducibility.

Table 11: Evaluation of related methods on VIKI-Bench.

| Method | VIKI-L1 (%) | VIKI-L2 (%) |
|---|---|---|
| GPT-4o | 18.40 | 17.50 |
| ReAct [52] | 18.70 | 19.88 |
| MAD (2 agents) [10] | **22.52** | **20.54** |

Table 12: Training resources for Qwen2.5-VL models on different task levels.

| Setting | Qwen2.5-VL-7B | | | Qwen2.5-VL-3B | | |
|---|---|---|---|---|---|---|
| | L1 | L2 | L3 | L1 | L2 | L3 |
| GPU Details | 8×A800 | 8×A800 | 8×A800 | 8×A800 | 8×A800 | 8×A800 |
| Training Time (h) | 22 | 9 | 5 | 13 | 5.5 | 3 |
| Training Epochs | 5 | 15 | 2 | 5 | 15 | 2 |
| Batch Size | 256 | 256 | 256 | 256 | 256 | 256 |

# I Prompt

**Output Constraint**

```
Output Format Requirements:
<answer>
  [
    {
      "step": 1,
      "actions": {'R1': ['Move', 'banana'], 'R2': ['Move', 'apple']}
    },
    {
      "step": 2,
      "actions": {'R1': ['Reach', 'banana'], 'R2': ['Reach', 'apple']}
    }
    # ... subsequent steps ...
  ]
</answer>
- step is the time step number (starting from 1, incrementing sequentially).
- Each robot can only have ONE action per time step.
- "actions" is a dictionary that specifies the action for each robot during a
↪   single time step. Each key (e.g., "R1", "R2") represents a robot. Each
↪   value is a list describing the single action that robot will perform in
↪   this step, with the following format: action_type,
↪   target_object_or_location, (optional: extra_argument)
Action primitives and descriptions: {ACTION_DESCRIPTION}
Available robot set: {robots}
Robot characteristics: {available_robots}
Their available operation APIs: {available_actions}
```

**Prompt of VIKI-L1**

```
Possible robots: {robot_set}
First, identify the robots visible in the image from a list of "possible
↪   robots". Among the visible robots, select the most suitable one or more
↪   to collaborate on the task.
```

Table 13: Hyperparameter settings for GRPO training

| Parameter | VIKI-L1 | VIKI-L2 | VIKI-L3 |
|---|---|---|---|
| Engine | $1:-vllm | $1:-vllm | $1:-vllm |
| VLLM attention backend | XFORMERS | XFORMERS | XFORMERS |
| Algorithm (adv estimator) | grpo | grpo | grpo |
| Train batch size | 256 | 256 | 256 |
| Max prompt length | 4096 | 4096 | 4096 |
| Max response length | 2048 | 2048 | 2048 |
| Filter overlong prompts | True | True | True |
| Truncation strategy | error | error | error |
| Actor learning rate | $1 \times 10^{-6}$ | $1 \times 10^{-6}$ | $1 \times 10^{-6}$ |
| Remove padding | True | True | True |
| PPO mini-batch size | 128 | 128 | 128 |
| PPO micro-batch per GPU | 10 | 10 | 10 |
| Use KL loss | True | True | True |
| KL loss coefficient | 0.01 | 0.01 | 0.01 |
| KL loss type | low_var_kl | low_var_kl | low_var_kl |
| Entropy coefficient | 0 | 0 | 0 |
| Gradient checkpointing | True | True | True |
| FSDP param offload | False | False | False |
| FSDP optimizer offload | False | False | False |
| Rollout log-prob micro-batch | 20 | 20 | 20 |
| Tensor model parallel size | 1 | 1 | 1 |
| Rollout engine name | vllm | vllm | vllm |
| GPU memory utilization | 0.6 | 0.6 | 0.6 |
| Chunked prefill | False | False | False |
| Enforce eager | False | False | False |
| Free cache engine | False | False | False |
| Rollout samples ($n$) | 5 | 5 | 5 |
| Reference log-prob micro-batch | 20 | 20 | 20 |
| Reference FSDP param offload | True | True | True |
| Use KL in reward | False | False | False |
| Critic warmup steps | 0 | 0 | 0 |
| GPUs per node | 8 | 8 | 8 |
| Number of nodes | 1 | 1 | 1 |
| epoch | 5 | 15 | 2 |

```
You FIRST think about the reasoning process as an internal monologue and then
↪   provide the final answer.
The reasoning process MUST BE enclosed within <think> </think> tags. The
↪   final answer MUST BE enclosed within <answer>Your final answer must be
↪   provided as a Python list format, for example: [\'fetch\',
↪   \'unitree_h1\']. Include only the robot names that are suitable for the
↪   task.</answer> tags.
```

**Prompt of VIKI-L2**

```
You are a plan creator. I will provide you with an image of robots in a
↪   scene, available robots and their action primitives, and a task
↪   description. You need to create a plan to complete the task.
You must first analyze the image to fully understand the scene depicted.
↪   Then, analyze the task description. Finally, create a plan to complete
↪   the task.
Your reasoning must strictly adhere to the visual content of the image and
↪   the task description-no assumptions, hypotheses, or guesses are allowed.
1. Create a plan to complete the task, noting:
    - Each robot can only perform ONE action per time step.
```

```
       - Multiple robots can work in parallel, but each robot is limited to one
       ↪  action at a time.
    2. You need to first provide your reasoning process within <think> and
    ↪  </think> tags.
    3. Your final answer must be within <answer> and </answer> tags, and
    ↪  **strictly follow the JSON format specified below**.

    Output Format Requirements(please comply strictly, do not output any
    ↪  additional content):
    <answer>
      [
        {{
          "step": 1,
          "actions": {{'R1': ['Move', 'banana'], 'R2': ['Move', 'apple']}}
        }},
        {{
          "step": 2,
          "actions": {{'R1': ['Reach', 'banana'], 'R2': ['Reach', 'apple']}}
        }}
        # ... subsequent steps ...
      ]
    </answer>
    Where:
    - step is the time step number (starting from 1, incrementing sequentially).
    - Each robot can only have ONE action per time step.
    - "actions" is a dictionary that specifies the action for each robot during a
    ↪  single time step. Each key (e.g., "R1", "R2") represents a robot. Each
    ↪  value is a list describing the single action that robot will perform in
    ↪  this step, with the following format: action_type,
    ↪  target_object_or_location, (optional: extra_argument)
    Action primitives and descriptions: {ACTION_DESCRIPTION}
    Available robot set: {robots}
    Robot characteristics: {available_robots}
    Their available operation APIs: {available_actions}
```

### Prompt of VIKI-L3

```
You are an expert in visual understanding and trajectory planning.
**INPUT:**
* An ego-view image showing two robotic arms working together; the arm
↪  closest to the camera represents **you**.
* A string describing the overall task.
* Two strings specifying your subtask ("you") and your partner's subtask.
**YOUR JOB:**
1. Enclose your scene analysis and task division within `<think>...</think>`
↪  tags.
2. Enclose your final output within `<answer>...</answer>` tags as a nested
↪  list of **ten 2D pixel coordinates**:
   * Two groups of five points each:
     * **First group:** your trajectory
     * **Second group:** your partner's trajectory
3. Follow this format **exactly** (no additional text):
   [[ [x1, y1], [x2, y2], [x3, y3], [x4, y4], [x5, y5] ],
    [ [x1', y1'], [x2', y2'], [x3', y3'], [x4', y4'], [x5', y5'] ]]
```

## Primitives and description

```
ROBOT_DESCRIPTION = {
    'stompy': 'A bipedal robot designed for dynamic walking and stomping
    ↪   tasks, featuring articulated arms. Color: Light blue body with yellow
    ↪   and orange accents.',
    'fetch': 'A wheeled robot with a flexible arm for object manipulation,
    ↪   designed for mobility and dexterity. Color: White with blue and black
    ↪   accents.',
    'unitree_h1': 'A humanoid robot with arms and legs designed for
    ↪   human-like movements and tasks. Color: Black.',
    'panda': 'A fixed robotic arm designed for precise and delicate
    ↪   manipulation tasks. Color: White with black accents.',
    'anymal_c': 'A quadrupedal robot built for navigating rough terrains and
    ↪   performing complex tasks with four articulated legs. Color: Red and
    ↪   black with some accents.',
    'unitree_go2': 'A compact quadrupedal robot optimized for agile movement
    ↪   and stability with four legs for efficient locomotion. Color: White.'
}
ACTION_DESCRIPTION = {
    'Move': "Command ['Move', 'object']: Robot R moves to the specified
    ↪   object.",
    'Open': "Command ['Open', 'object']: Open the object held by the Robot
    ↪   R's end effector.",
    'Close': "Command ['Close', 'object']: Close the object held by the Robot
    ↪   R's end effector.",
    'Reach': "Command ['Reach', 'object']: Robot R reaches the specified
    ↪   object.",
    'Grasp': "Command ['Grasp', 'object']: Robot R's end effector performs a
    ↪   grasping operation on a specified object.",
    'Place': "Command ['Place', 'object']: Place the object held by the Robot
    ↪   R's end effector at a specified location (the release point, not the
    ↪   object itself).",
    'Push': "Command ['Push', 'object', 'R1']: Robot R pushes the object to
    ↪   robot R1.",
    'Interact': "Command ['Interact', 'object']: A general interaction
    ↪   operation, flexible for representing interactions with any asset."

}
AGENT_AVAIL_ACTIONS = {
    'panda':      ['Reach', 'Grasp', 'Place', 'Open', 'Close', 'Interact'],
    'fetch':      ['Move', 'Reach', 'Grasp', 'Place', 'Open', 'Close',
    ↪   'Interact'],
    'unitree_go2':['Move', 'Push', 'Interact'],
    'unitree_h1': ['Move', 'Reach', 'Grasp', 'Place', 'Open', 'Close',
    ↪   'Interact'],
    'stompy':     ['Move', 'Reach', 'Grasp', 'Place', 'Open', 'Close',
    ↪   'Interact'],
    'anymal_c':   ['Move', 'Push', 'Interact'],
}

AGENT_END_EFFECTOR_NUM = {
    'panda': 1,
    'fetch': 1,
    'unitree_go2': 0,
    'unitree_h1': 2,
    'stompy': 2,
    'anymal_c': 0,
}
```

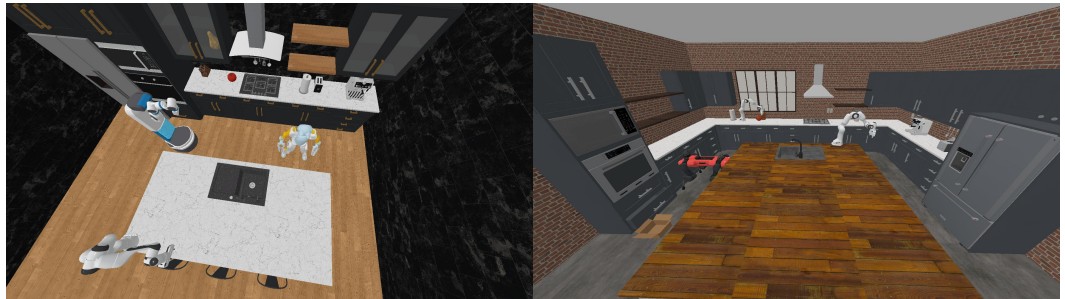

**Task Description:** Inspect the table visually. Ensure both the **pear** and the **tomato** are present. Fetch and place the missing fruit, checking all areas including cabinet.

**Agent Activation:** R1: fetch, R2: stompy

**Task Planning:**
Step 1: <R1, Move tomato>, <R2, Move cabinet>
Step 2: <R1, Reach tomato>, <R2, Reach cabinet>
Step 3: <R1, Grasp tomato>, <R2, Open cabinet>
Step 4: <R1, Move table>, <R2, Move pear>
Step 5: <R1, Place table>, <R2, Reach pear>
Step 6: <R2, Grasp pear>
Step 7: <R2, Move table>
Step 8: <R2, Place table>

**Task Description:** Get the **peach** over to the sink. In case it's too distant, put the peach into a **cardboardbox**, shift the cardboardbox to the sink, retrieve the peach.

**Agent Activation:** R1: panda, R2: panda, R3: anymal_c

**Task Planning:**
Step 1: <R2, Reach peach>, <R3, Move cardboardbox>
Step 2: <R2, Grasp peach>, <R3, Push cardboardbox R2>
Step 3: <R2, Place cardboardbox>
Step 4: <R3, Push cardboardbox R1>
Step 5: <R1, Reach peach>
Step 6: <R1, Grasp peach>
Step 7: <R1, Place sink>

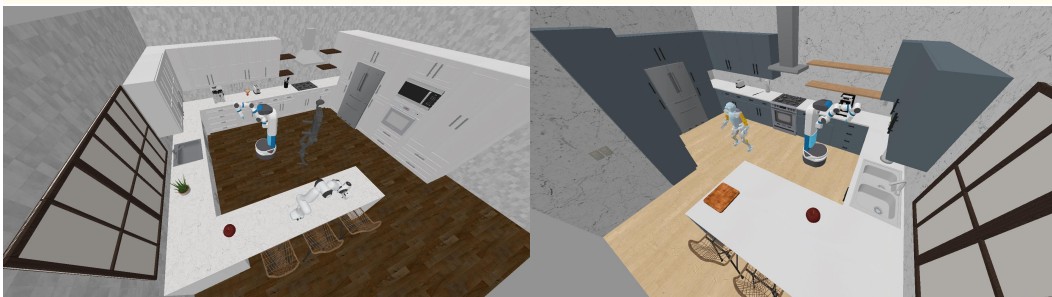

**Task Description:** Move both the **meat** and the **apple** into the **bowl** to prepare for serving.

**Agent Activation:** R1: fetch, R2: unitree_h1

**Task Planning:**
Step 1: <R1, Move apple>, <R2, Move meat>
Step 2: <R1, Reach apple>, <R2, Reach meat>
Step 3: <R1, Grasp apple>, <R2, Grasp meat>
Step 4: <R1, Move bowl>, <R2, Move bowl>
Step 5: <R1, Place bowl>, <R2, Place bowl>

**Task Description:** Position the **apple** on the **wooden cutting board;** slice it neatly with the **knife**.

**Agent Activation:** R1: stompy, R2: fetch

**Task Planning:**
Step 1: <R1, Move tomato>, <R2, Move cabinet>
Step 2: <R1, Reach tomato>, <R2, Reach cabinet>
Step 3: <R1, Grasp tomato>, <R2, Open cabinet>
Step 4: <R1, Move table>, <R2, Move pear>
Step 5: <R1, Place table>, <R2, Reach pear>
Step 6: <R2, Grasp pear>
Step 7: <R2, Move table>
Step 8: <R2, Place table>

Figure 8: Qualitative demonstrations of VIKI-L1 and VIKI-L2 showcasing visual input, agent activation, and task planning across diverse multi-agent collaboration scenarios.

