# OpenReview forum: "VIKI‑R: Coordinating Embodied Multi-Agent Cooperation via Reinforcement Learning"
_NeurIPS.cc/2025/Datasets_and_Benchmarks_Track — NeurIPS 2025 Datasets and Benchmarks Track poster_

### Official Review · Reviewer_mtDG · 2025-06-30

**Rating:** 4
**Confidence:** 3

**Summary:**

They introduce VIKI-Bench, the first hierarchical benchmark tailored for embodied multi-agent cooperation, featuring three structured levels: agent activation, task planning, and trajectory perception. VIKI-Bench includes diverse robot embodiments, multi-view visual observations, and structured supervision signals to evaluate reasoning grounded in visual inputs. To demonstrate the utility of VIKI-Bench, they propose VIKI-R, a two-stage framework that fine-tunes a pretrained vision-language model (VLM) using Chain-of-Thought annotated demonstrations, followed by reinforcement learning under multi-level reward signals.

**Dataset Code Accessibility:**

Yes

**Ethical Considerations:**

No, there are no or only very minor ethics concerns

**Final Justification:**

The authors have addressed most of my concerns.

**Limitations Weaknesses:**

- As a benchmark paper, the selection of models or algorithms evaluated is relatively limited, focusing primarily on ablated versions of the proposed VIKI-R algorithm. It would strengthen the paper to include comparisons with other algorithms developed for vision-based multi-agent coordination. If there are currently no related works, the authors should further clarify the significance and necessity of introducing this benchmark at this stage.
- Additional experiments demonstrating the diversity and customizability of the proposed benchmark would be beneficial.
- The paper would be improved by providing more comprehensive ablation studies, including evaluations with different reward designs, RL-tuning algorithms, prompt templates, and model sizes.
- More visualizations showcasing demos of the benchmark would help readers better understand its functionality and potential applications.
- Including tutorials on using and extending the benchmark would be valuable for the community and facilitate wider adoption.

**Strengths Contributions:**

- This paper proposes a benchmark for evaluating algorithms designed to reason about visual observations in heterogeneous multi-agent settings. As shown in Table 1, this benchmark addresses an important research gap in the field.
- In addition, the authors present an algorithm that demonstrates improved performance in these settings, achieving superior results even with a smaller model size.

---

> ### Author Rebuttal · Authors · 2025-07-30
>
> We thank the reviewer for their insightful observations and valuable suggestions. We appreciate the acknowledgment of our benchmark and performance improvements of our proposed algorithm. In this response, we provide further clarifications and analyses to address the concerns and strengthen our contributions.
>
> ### Q1: Extension of Evaluation to Other Relevant Methods
>
> A: Currently, there are no existing methods that are exactly aligned with the setting of our proposed benchmark, which focuses on vision-based multi-agent coordination in hierarchical tasks. However, we have adapted and evaluated several relevant approaches developed for vision-based multi-agent collaboration, including [1][2]. These methods were adapted to fit our benchmark’s protocol and input-output format. The evaluation results are summarized as follows:
>
> | Method             | VIKI-L1 | VIKI-L2 |
> |--------------------|---------|---------|
> | GPT-4o             | 18.40   | 17.50   |
> | React [1]          | 18.70   | 19.88   |
> | MAD (2 agents) [2] | 22.52   | 20.54   |
>
> Moreover, the significance of our work and benchmark lies in the growing trend of multiple robots cooperation in multi-agent systems. However, current datasets lack the content to evaluate such multi-agent coordination in vision-based tasks, while our work fills this critical gap by providing a progressive approach to enhancing the fundamental capabilities of multi-agent systems. Additionally, we introduce several strategies based on fine-tuning and reinforcement learning within our proposed VIKI-R algorithm. This highlights the effectiveness of our method and serves as a foundation for future developments in multi-agent vision-based coordination.
>
> [1] ReAct: Synergizing Reasoning and Acting in Language Models
>
> [2] Improving factuality and reasoning in language models through multiagent debate
>
> ### Q2: Significance of VIKI-Bench
>
> A: VIKI-Bench demonstrates significance for evaluating multi-agent cooperation through a range of challenging tasks that test capabilities like agent recognition, planning and control.
>
> First, The hierarchical task levels require agents to reason across multiple levels to ensure effective cooperation. This complexity increases as agents interact with each other and reason through different tasks. In addition, the tasks also exhibit significant data diversity. The number of agents directly impacts difficulty, with more robots adding complexity to coordination. Tasks involving more steps or robots increase coordination and reasoning demands. Moreover, the tasks require the planner to recognize agents and assign roles, as well as understand spatial relationships for concurrent planning and trajectory prediction.
>
> These features make VIKI-Bench a valuable benchmark for evaluating the complex abilities needed for multi-agent coordination and cooperation.
>
> ### Q3: Demonstrations of Diversity
>
> A: We will include visualizations in the final version of the paper to illustrate the diversity of tasks, environments, and agent configurations in VIKI-Bench. These visual examples will highlight variations in scene layouts, object types, robot embodiments, and task goals, providing an intuitive understanding of the benchmark’s compositional richness and coverage.
>
> ### Q4: Ablation Stuides
>
> A: We conducted a series of ablation studies to analyze the effectiveness and design choices of VIKI-R from multiple perspectives, including reward structure, model scaling, prompt formulation, and policy optimization strategies. The experiments below provide detailed comparisons that validate the rationality and robustness of our framework.
>
> 1. Comparison of Different Reward Settings
>
> For the VIKI-2 dataset, we conducted an ablation study to investigate the impact of removing the step reward. The results are shown below. We observe that without the step reward, both in-distribution (ID) and out-of-distribution (OOD) accuracies drop, highlighting the importance of the reward design in VIKI-R.
> | Model                               | VIKI-L2-ID | VIKI-L2-OOD |
> |-------------------------------------|------------|-------------|
> | Qwen2.5-VL-7B-Inst (with step)      | 95.22      | 33.25       |
> | Qwen2.5-VL-7B-Inst (without step)   | 92.06      | 29.51       |
>
> 2. Impact of Model Scale
>
> We further introduced a larger-scale model, Qwen2.5-VL-32B-Inst, for comparison. The results demonstrate that increasing model scale significantly improves performance, indicating that scaling helps the model solve complex reasoning problems more effectively.
>
> | Model                  | VIKI-L1 | VIKI-L2 |
> |------------------------|---------|---------|
> | Qwen2.5-VL-3B-Inst     | 74.10   | 68.78   |
> | Qwen2.5-VL-7B-Inst     | 93.00   | 69.25   |
> | Qwen2.5-VL-32B-Inst    | 95.30   | 74.74   |
>
> 3. Effect of Prompt Design on Training
>
> On the VIKI-1 dataset, we modified the prompt by removing reasoning guidance such as:
> "You FIRST think about the reasoning process as an internal monologue and then provide the final answer."
> We found that while accuracy remained largely unchanged, the training process took longer to converge. This suggests that reinforcement learning can effectively explore and discover reasoning paths even without explicit guidance.
>
> | Model                                 | VIKI-L1 Accuracy |
> |--------------------------------------|------------------|
> | Qwen2.5-VL-3B-Inst                    | 74.10            |
> | Qwen2.5-VL-3B-Inst (w/o guidance)     | 73.47            |
> | Qwen2.5-VL-7B-Inst                    | 93.00            |
> | Qwen2.5-VL-7B-Inst (w/o guidance)     | 92.95            |
>
> 4. Comparison Between DAPO and GRPO
>
> We also conducted ablation studies between DAPO [3] and GRPO. The results show that DAPO achieves similar accuracy, indicating that our approach is robust to different policy optimization methods.
> | Model                         | VIKI-L2-ID | VIKI-L2-OOD |
> |-------------------------------|------------|-------------|
> | Qwen2.5-VL-3B-Inst            | 93.61      | 32.11       |
> | Qwen2.5-VL-3B-Inst (DAPO)     | 93.22      | 33.33       |
> | Qwen2.5-VL-7B-Inst            | 95.22      | 33.25       |
> | Qwen2.5-VL-7B-Inst (DAPO)     | 95.44      | 33.58       |
>
> [3] DAPO: An Open-Source LLM Reinforcement Learning System at Scale
>
> ### Q5: Visualizations on benchmark
>
> A: We have included sample visualizations of our benchmark in Supplementary E. A more comprehensive set of demonstrations will be available in the official version.
>
> ### Q6: Tutorials
>
> A: We have provided comprehensive experimental configurations and reproduction parameters in Section C.3 and D in the supplementary. We will provide detailed tutorials in the official repository, covering the usage of the benchmark as well as instructions on customizing configurations for a wide range of agents, scenes, and tasks.

---

> > ### Comment · Reviewer_mtDG · 2025-08-03
> >
> > I thank the authors for the detailed response.
> >
> > My concerns are generally addressed, and I will raise my score to 4.

---

### Official Review · Reviewer_dVhv · 2025-07-01

**Rating:** 6
**Confidence:** 4

**Summary:**

The paper introduces VIKI-Bench, a hierarchical benchmark for evaluation vision language models (VLMs) in embodied multi-agent cooperative settings and VIKI-R, a two-stage framework to enhance VLM reasoning capabilities in this context. The dataset consists a varied set of scenarios for future work.

**Additional Feedback:**

I had a few questions for the authors:

1. Given the dependence of GRPO on the base policy quality, have the authors explored strategies to improve the initial performance of the base policy, especially for complex tasks like VIKI-L2 planning, to mitigate the “cold start” issue? For instance, could more targeted pre-training or alternative initial supervised fine-tuning strategies be beneficial?
2. The paper mentions that reinforcement learning enables the emergence of compositional cooperation patterns. Can the authors elaborate on the nature of these emergent patterns? For example, are they explicitly interpretable sequences of actions, or more implicit, learned coordination strategies? Is there a mechanism to transfer or generalize these emergent patterns to new, unseen tasks or agent combinations?
3. The task planning reward is defined to favor efficient solutions (plan length not exceeding ground-truth length). While efficiency is important, are there scenarios where a slightly longer, but more robust or safer, plan might be preferred? How could the reward function be extended to incorporate such qualitative aspects of plan quality?
4. What are the computational resources (e.g., specific GPUs, training time) required for fine-tuning the 7B and 3B parameter Qwen2.5-VL models using VIKI-R? Providing these details would be beneficial for other researchers looking to reproduce or build upon this work (especially in this track at NeurIPS)
5. Could the iterative refinement process for task planning, currently using GPT-4o as the plan generator, be effectively integrated within the VIKI-R framework itself, potentially using the fine-tuned VLM for self-correction and refinement without external LLM calls during inference?

1. Does VIKI-Bench allow for customisation with object/agent configuration? If yes, could the authors provide more information either in the paper/supplementary material or the codebase

**Dataset Code Accessibility:**

Yes

**Ethical Considerations:**

No, there are no or only very minor ethics concerns

**Final Justification:**

Thank you for addressing my concerns. I am assuming that these changes will be added to the main paper (related work, appendix, etc.). I would like to increase my score by 1 since the authors did a good job with answering my questions.

**Limitations Weaknesses:**

1. I feel that the "plan-generator+task-allocation+verification+refinement" approach is quite similar to [1]. I believe that the major difference is that the number of scenarios in VIKI-Bench is way more than in [1]. I think a comparison/noting the differences to this work would be good in table 1.
2. The iterative refinement process for Task Planning relies on an external "Plan Generator $g_{plan}$" (GPT-4o). While effective, this suggests that the full planning capability of VIKI-Bench might depend on an external, potentially closed-source and computationally expensive Large Language Model, rather than being fully self-contained within the VLM fine-tuned by VIKI-R. Could you please provide an estimate of the costs of all the queries made to GPT-4o? And how would the performance vary with varying the VLM for this?
3. While I do not expect the current framework to solve this completely (I understand that the current work is a step towards this, I would appreciate some comments on challenges to extend to a real-world setting and what are the current limitations of the VIKI framework. Since this is a common but important limitation, challenges like real-world sensor noise, latency, and physical interaction complexities are not fully addressed/mentioned.


References:

[1]: Nayak, S., Morrison Orozco, A., Have, M., Zhang, J., Thirumalai, V., Chen, D., ... & Balakrishnan, H. (2024). Long-horizon planning for multi-agent robots in partially observable environments. *Advances in Neural Information Processing Systems*, *37*, 67929-67967.

**Strengths Contributions:**

* VIKI-Bench is presented as the first hierarchical benchmark tailored for embodied multi-agent cooperation, featuring three structured levels: agent activation, task planning, and trajectory perception. This multi-level approach is crucial for supporting fine-grained teamwork and comprehensive evaluation and could potentially be used by a lot of researchers for advancing multi-agent embodied robotic applications.
* Embodiment Diversity and Visual Grounding: The benchmark includes diverse robot embodiments and multi-view visual observations, allowing for evaluation of reasoning grounded in visual inputs. This is a significant improvement over existing VLM-based approaches that are limited in their support for diverse settings.
* Effective Two-Stage Framework (VIKI-R): The proposed VIKI-R framework, which fine-tunes a pretrained VLM using Chain-of-Thought (CoT) annotated demonstrations followed by reinforcement learning (RL) under multi-level reward signals, demonstrates significant outperformance over baseline methods across all task levels. The integration of RL is shown to substantially enhance visual reasoning capabilities in hierarchical multi-agent cooperation.
* Structured Data Generation: The paper details a robust data generation process for each level of VIKI-Bench, including visual reasoning for agent activation, an iterative refinement process with GPT-4 for task planning, and spatial keypoint prediction for trajectory perception. The use of GPT-40 as a task allocator and plan generator, combined with verification and refinement modules, contributes to high-quality data.
* Comprehensive Experimental Setup: The experiments compare VIKI-R against various open-source and closed-source models (GPT-4o, Gemini-2.5-Flash-preview, Claude-3.7-Sonnet) and different training paradigms, including Ans-SFT and VIKI-R-Zero. This thorough comparison strengthens the claims of VIKI-R's effectiveness. I appreciate the details provided in the supplementary material as well.
* Detailed Ablation Studies and Insights: The paper provides valuable insights from training, including the critical dependence of GRPO on base policy quality and the evolution of response length during training. The ablation study on step penalty clearly demonstrates its positive impact on generalization and execution accuracy.
* Reproducibility: The authors have provided access to the data and code with sufficient instructions for faithful reproduction of the main experimental results.

---

> ### Author Rebuttal · Authors · 2025-07-30
>
> We express our sincere gratitude to the reviewer for their detailed and constructive comments. We appreciate the recognition of the VIKI-Bench benchmark and the effectiveness of the VIKI-R framework. We offer additional clarifications and analyses on our work in the following response.
>
> ### Q1: Comparsion with current benchmarks (LLaMAR)
> A: We provide a comprehensive comparison with the MAP-THOR benchmark in LLaMAR. As shown in table below, LLaMAR (MAP-THOR) focuses on multi-agent task planning and refinement, with 225 task instances, emphasizing task planning and verification. VIKI-Bench goes a step further by offering a broader evaluation framework, featuring over 23,000 task instances across 100 diverse scenes and multiple robot types. Both egocentric and global views provides a more comprehensive testing environment for multi-agent systems, significantly advancing their cooperation, scalability, and adaptability in real-world settings.
>
> | Environment           | Language | Visual | Views        | H.E. | Tasks Num     |
> |-----------------------|----------|--------|--------------|------|----------------|
> | LLaMAR (MAP-THOR) [1] | 3D       | ✓      | ✓ (EGO, GL)  | ✓    | 225  |
> | VIKI-Bench (Ours)     | 3D       | ✓      | ✓ (EGO, GL)  | ✓    |  23,737  |
>
> [1]: Nayak, S., Morrison Orozco, A., Have, M., Zhang, J., Thirumalai, V., Chen, D., ... & Balakrishnan, H. (2024). Long-horizon planning for multi-agent robots in partially observable environments. Advances in Neural Information Processing Systems, 37, 67929-67967.
>
> ### Q2: Costs of External VLMs and performance comparisons
>
> A: The iterative refinement process is used during the data generation process of VIKI-Bench, where we employ GPT-4o via API calls to generate task plans, rather than the inference stage. On average, generating one task costs approximately $0.05 using GPT-4o. We have experimented with other models, including the open-source Qwen2.5-VL-72B-Instruct, but observed significantly inferior performance. Specifically, Qwen-based models often require more than 20 rounds of reasoning to generate a valid and coherent plan, leading to both inefficiency and instability. In contrast, GPT-4o demonstrates stronger generalization and reliability in complex multi-agent planning scenarios, making it a more practical choice for dataset construction.
>
> ### Q3: Limitations and Extension of VIKI-R
>
> A: The main limitation of our work is that it is conducted in simulation due to the high costs associated with hardware consumption and repairs when using multiple heterogeneous robots. While our primary experiments are performed in simulation (ManiSkill) to reduce hardware costs and ensure safety, we have also designed and implemented real-world settings to assess feasibility. We instantiate 100 VIKI-L1 tasks using GPT-4o to generate instructions and visual observations tailored to the real-world scenes, and evaluate the zero-shot performance of several leading vision-language models. Preliminary results from the real-world VIKI-L1 setting are summarized below:
>
> | Model Name              | VIKI-L1  |
> |-------------------------|----------|
> | Qwen2.5-VL-72B-Instruct | 14%      |
> | Qwen2.5-VL-32B-Instruct | 8%       |
> | Claude-3-7-Sonnet       | 15%      |
> | Gemini-2.5-Flash        | 28%      |
>
> We also design feasible real-world setups for VIKI-L2 and VIKI-L3, though full-scale deployment is ongoing due to hardware complexity. Concretely, we design real-world tasks aligned with the hierarchical structure of VIKI-Bench, and collect multi-view visual observations of physical robots across diverse environments (e.g., lab desktops, industrial setups). Following the same formulation as in simulation, we aim to construct datasets under the VIKI-L2 and L3 settings, enabling evaluation of reasoning and spatial consistency in real-world scenarios.
>
> These efforts highlight the performance gap between simulation and reality, and confirm that real-world evaluation is both meaningful and challenging. We see this as a promising direction for future expansion.
>
> We agree that expanding the evaluation to real-world testing is a promising direction for future work, as it would provide valuable insights into the robustness and applicability of current methods.
>
> ### Q4: Strategies for Cold-start
> A: While task-specific pretraining (e.g., mixing reasoning data with large VQA datasets) has shown potential to enhance model performance, especially in terms of improving the initial policy for reinforcement learning, we have not incorporated this step in our current work due to the significant computational and engineering overhead it entails. Nevertheless, existing literature suggests that such pretraining could be beneficial and may be considered in future iterations. Instead, we focus on CoT-based supervised fine-tuning, which offers a more practical trade-off and still provides meaningful gains in early-stage performance and convergence speed in complex reasoning tasks.
>
> ### Q5: Elaboration on Compositional Cooperation Patterns
> A: In our work, Reinforcement Learning (RL) improves the success rate by enabling the emergence of compositional cooperation patterns, which is both explicit and implicit. Through CoT-based fine-tuning and RL, the model learns stronger reasoning capabilities, allowing it to discover explicit patterns, such as the temporal order in multi-agent planning tasks. At the same time, more implicit patterns like coordinated actions in trajectory prediction emerge. These learned patterns are core to effective multi-agent cooperation.
>
> In addition, RL enhances the model’s ability to generalize, allowing previously learned tasks to transfer to new, unseen scenes or agent combinations. As demonstrated in Tab. 2, Qwen2.5VL-7B-Instruct trained on VIKI-L2 in an In-Domain setting maintain relatively high performance when applied to Out-of-Domain tasks, showcasing the model’s adaptability to new environments.
>
> ### Q6: Extension on Safety
>
> A: In safety-sensitive scenarios, agents can prioritize robustness over minimal plan length with multiple methods. First, the reward function can be extended to incorporate qualitative aspects of plan stabilities by integrating a safety score for factors like speed, stability, and proximity to other agents, ensuring safer operations even at the cost of reduced efficiency. Additionally, agents can employ proactive detection mechanisms, such as pre-action verification steps or environmental checks, to mitigate risks before executing critical actions. Techniques like curriculum learning can further enhance safety by progressively training agents in increasingly complex environments, gradually building their capability to handle complex situations while maintaining efficiency.
>
> ### Q7: Computation Resources
>
> A: We will provide detailed training times and resource usage used during fine-tuning.
>
> Experiment in Qwen2.5-VL-7B:
> |                   | VIKI-L1           | VIKI-L2           | VIKI-L3           |
> |-------------------|-------------------|-------------------|-------------------|
> | GPU Details       | 8 × NVIDIA A800   | 8 × NVIDIA A800   | 8 × NVIDIA A800   |
> | Training Time     | 22h               | 9h                | 5h                |
> | Training Epochs   | 5                 | 15                | 2                 |
> | Batch Size        | 256               | 256               | 256               |
>
> Experiment in Qwen2.5-VL-3B:
> |                   | VIKI-L1           | VIKI-L2           | VIKI-L3           |
> |-------------------|-------------------|-------------------|-------------------|
> | GPU Details       | 8 × NVIDIA A800   | 8 × NVIDIA A800   | 8 × NVIDIA A800   |
> | Training Time     | 13h               | 5.5h              | 3h                |
> | Training Epochs   | 5                 | 15                | 2                 |
> | Batch Size        | 256               | 256               | 256               |
>
> ### Q8: Dependence on External LLMs in Planning
>
> A: Our approach leverages iterative refinement process in both planning data generation and inference, but with distinct implementations. During data generation, we employ GPT-4o to synthesize high-quality reasoning traces across diverse tasks. At inference time, however, our method relies solely on the fine-tuned backbone model to refine plans through self-correction, without the introduction of external LLMs. Notably, after fine-tuning on VIKI-Bench, the backbone achieves performance competitive with GPT-4o, demonstrating that the iterative refinement capability can be effectively integrated with the VIKI-R framework while maintaining effectiveness.
>
> ### Q9: Customization for Configuration
>
> A: We offer customization for various aspects of the configuration, including the number and type of agents, the selection of assets, as well as the type of tasks and scenes. We ensure flexibility for users to tailor the benchmark to specific experimental needs and scenarios. The detailed information and tutorials will be provided in the official repository.

---

> > ### Comment · Area_Chair_yASs · 2025-08-04
> > **Please engage with the authors' rebuttal**
> >
> > Hi Reviewer dVhv,
> >
> > Thanks for your detailed review. The authors have provided a rebuttal addressing your points. Since there are only a couple days remaining for the discussion period with the authors, please engage with their rebuttal as soon as possible.
> >
> > Cheers,
> > Your AC

---

### Official Review · Reviewer_Ktr9 · 2025-07-03

**Rating:** 4
**Confidence:** 3

**Summary:**

This paper introduces VIKI-Bench, a benchmark for evaluating VLM in embodied multi-agent cooperation scenarios, along with VIKI-R, a two-stage learning framework. The benchmark features three levels of tasks: agent activation, task planning, and trajectory perception. The framework combines supervised fine-tuning with reinforcement learning to enhance visual reasoning capabilities. The authors demonstrate that their approach outperforms baseline methods and closed-source models across various metrics.

**Dataset Code Accessibility:**

Yes

**Dataset Code Comments:**

Code is provided

**Ethical Considerations:**

No, there are no or only very minor ethics concerns

**Final Justification:**

All issues resolved.

**Limitations Weaknesses:**

Where is the difficulty of the tasks come from? Is it from the number of agents (more agents will be more difficult to coordinate)? Or the task level (how many steps are needed to accomplish each task)? Or from the vision input? There is missing of ablation experiments to figure those out.

What is the success rate of VIKI-R?
In table 2 we see high numbers on VIKI-L1. What is the room of improvement for new methods? What should be the directions of the newly developed algorithm?

What is the analysis between the VIKI-L1 -VIKI-L2- VIKI-L3?

**Strengths Contributions:**

1. benchmark specifically designed for embodied multi-agent cooperation with visual reasoning. It has large diversities and complexities.

2. The proposed method shows clear improvements over baselines.

3. The data curation method is detailed and could be extended to new tasks generation

---

> ### Author Rebuttal · Authors · 2025-07-30
>
> We express our sincere gratitude to the reviewer's constructive comments. We appreciate the recognition of the novelty of the VIKI-Bench benchmark and the VIKI-R framework. We provide additional analyses and insights in response to the raised concerns in the review.
>
> ### Q1: Justifications of task difficulties
>
> A: The difficulties of our tasks arise from several aspects.
>
> (1) The hierarchical task levels emphasize the need for multi-agent cooperation. These tasks demand a broad range of abilities, including both planning and control. The complexity grows as agents not only interact with each other but also reason across different task levels to ensure effective cooperation.
>
> (2) The tasks exhibit considerable data diversity in both structural and reasoning aspects, significantly impacting task difficulty. One key factor is the number of agents involved. As the number of robots increases, the coordination complexity grows substantially. For example, as shown in the VIKI-2 dataset, task accuracy drops from 98.43% with a single robot to 94.53% with two robots, and further to 43.75% with three robots. This sharp decline highlights the increased challenge of multi-agent coordination and communication.
>
> Another important dimension of task diversity is the reasoning horizon. Tasks that require more steps inherently demand deeper planning and long-term consistency. Our results show that tasks with 3 or fewer steps achieve 100% accuracy, whereas accuracy decreases to 96.21% for 4–6 steps, and further to 93.26% for 7–9 steps. This trend underscores the difficulty of maintaining coherent long-horizon reasoning, especially when combined with the need for coordination across multiple agents.
> Together, these observations illustrate that both the number of agents and the planning depth independently and jointly contribute to the overall complexity of the tasks. Effective policies must therefore be capable of handling multi-agent coordination and long-horizon reasoning simultaneously—an aspect that makes this benchmark particularly challenging and representative of real-world scenarios.
>
> | Steps Range | VIKI-L2 Accuracy | Robots | VIKI-L2 Accuracy |
> |---------------|------------------|----------|------------------|
> | ≤ 3           | 100%             | 1        | 98.43%           |
> | 4–6           | 96.21%           | 2        | 94.53%           |
> | 7–9           | 93.26%           | 3        | 43.75%           |
>
> (3) The tasks demand a comprehensive set of capabilities, including agent recognition, assignment, and spatial perception. These elements are crucial for concurrent planning and trajectory prediction, as agents must navigate the environment while considering their relative positions and task assignments.
>
> ### Q2: VIKI-R success rates and future directions for improving performance (include VIKI-L1)
>
> Table 2 report the success rates of VIKI-R (see row "+VIKI-R"). VIKI-R achieves success rates of 93.00% on VIKI-L1, 69.25% on VIKI-L2, and average distance 77.82 on VIKI-L3, as reported in Table 2. While the performance on VIKI-L1 appears near-saturated, there remains substantial room for improvement on the more challenging VIKI-L2 and VIKI-L3 levels. Further research can explore several promising directions: (1) leveraging more effective supervised learning strategies, such as task-specific pretraining and high-quality annotations, to improve initial model capabilities; (2) incorporating reinforcement learning to enhance generalization and decision-making in complex tasks; and (3) designing structured multi-agent collaboration frameworks that make full use of the diversity within the VIKI dataset to build and validate embodied agents. These advances will enable models to better address the core challenges of multi-agent embodied intelligence.
>
> ### Q3: Analysis between VIKI-L1, L2, L3
>
> A: The relationship between VIKI-L1, L2, L3 focuses on the increasing complexity and specialization of each task, which is introduced in Section 3.1 and 3.2 of the paper.
>
> (1) VIKI-I is designed for agent allocation with an emphasis on spatial understanding, evaluating the model's ability to recognize and assign agents within an environment. VIKI-II involves multi-agent planning, targeting long horizontal reasoning and grounding, where the model plan and coordinate multiple robots in the environments. VIKI-III focuses on multi-agent trajectory prediction, specifically from the egocentric views of agents, evaluating the capability to predict the movement of agents in dynamic environments.
>
> (2) These three hierarchical tasks together evaluate the comprehensive capabilities of models in multimodal understanding and spatial perception, which are essential for effective multi-agent coordination and cooperation. Each task progressively builds on the previous one, assessing how well the model handles increasing levels of complexity and coordination required for real-world multi-agent scenarios.

---

> > ### Comment · Area_Chair_yASs · 2025-08-04
> > **Please engage with the authors' rebuttal**
> >
> > Hi Reviewer Ktr9,
> >
> > The authors have provided a rebuttal addressing your points. Since there are only a couple days remaining for the discussion period with the authors, please engage with their rebuttal as soon as possible.
> >
> > Cheers,
> > Your AC

---

> > ### Comment · Reviewer_Ktr9 · 2025-08-05
> >
> > Thanks for the answer. it addressed my concern and I will raise my score.

---

### Official Review · Reviewer_UsEP · 2025-07-06

**Rating:** 5
**Confidence:** 4

**Summary:**

This paper presents a hierarchical benchmark designed to evaluate vision-language models (VLMs) in the context of embodied multi-agent cooperation. The benchmark is structured around three level tasks: (1) agent activation, (2) task planning and (3) trajectory perception.  To address these task, a two-stage learning framework that combines SFT with CoT annotations and GRPO.
The paper presents experiments on multiple models, including GPT-4o and Qwen variants and it shows that the framework significantly outperforms baseline and zero-shot methods across all three levels, especially in compositional cooperation tasks.

**Dataset Code Accessibility:**

Yes

**Dataset Code Comments:**

- All task components (agent types, image views, primitives, constraints) are fully documented.

- CoT formats are standarized using <think> ...</think> and <answer> ..</answer> tags for consistency.

-Full data format and annotation procedures are provided.

-The paper provides training objectives and algorithms.

**Ethical Considerations:**

No, there are no or only very minor ethics concerns

**Final Justification:**

The new experiments demonstrate the practicality of the benchmark, but also highlight key gaps between current model performance and human capabilities.

**Limitations Weaknesses:**

- All tasks are evaluated in simulation; no deployment or test in real-world robot environments is shown.

-Smaller-scale open-source models underperform significantly compared to their 72B counterpart and commercial models, suggesting high compute dependency.

-Evaluation lacks comparison with human experts or human-agent interaction.

-Without prior CoT grounding RL finetuning leads to poor performance.

**Strengths Contributions:**

- It claims it is the first hierarchical benchmark targeting embodied multi-agent visual reasoning, supporting heterogeneous robot types, multi-view inputs (global + egocentric), and realistic task structures.
-The benchmark includes agent selection, action sequence generation and trajectory prediction.
-It is evaluated on open-source and closed-source VLMS.
-The proposed framework outperforms both supervised-only and reinforcement-only baselines on all sub-tasks.
-Reinforcement learning notably improves generalization to out-of-domain tasks.

---

> ### Author Rebuttal · Authors · 2025-07-30
>
> We sincerely thank the reviewer for the thoughtful feedback. We greatly appreciate the positive remarks regarding the hierarchical task design, comprehensive evaluations, and the promising impact of our work. In this rebuttal, we provide further clarifications and additional experiments to address the concerns raised in the review.
>
> ### Q1: Evaluation under Real-world Deployment
>
> A: It is indeed reasonable to conduct real-world evaluations for tasks in VIKI-Bench. While our primary experiments are performed in simulation (ManiSkill) to reduce hardware costs and ensure safety, we have also designed and implemented real-world settings to assess feasibility. Specifically, we deploy a mobile dual-arm RealMan robot, two single-arm Panda robots, and a dual-arm Agilex robot across physical environments.
>
> We instantiate 100 VIKI-L1 tasks using GPT-4o to generate instructions and visual observations tailored to the real-world scenes, and evaluate the zero-shot performance of several leading vision-language models. Preliminary results from the real-world VIKI-L1 setting are summarized below:
>
> **Tab. R1: Real-world performance**
>
> | Model Name              | VIKI-L1  |
> |------------------------------|----------|
> | Qwen2.5-VL-72B-Instruct | 14%      |
> | Qwen2.5-VL-32B-Instruct | 8%       |
> | Claude-3-7-Sonnet       | 15%      |
> | Gemini-2.5-Flash        | 28%      |
>
> We also design feasible real-world setups for VIKI-L2 and VIKI-L3, though full-scale deployment is ongoing due to hardware complexity. Concretely, we design real-world tasks aligned with the hierarchical structure of VIKI-Bench, and collect multi-view visual observations of physical robots across diverse environments (e.g., lab desktops, industrial setups). Following the same formulation as in simulation, we aim to construct datasets under the VIKI-L2 and L3 settings, enabling evaluation of reasoning and spatial consistency in real-world scenarios.
>
> These efforts highlight the performance gap between simulation and reality, and confirm that real-world evaluation is both meaningful and challenging. We see this as a promising direction for future expansion.
>
> ### Q2: High Computational Dependency
>
> A: It is common that large-scale models typically outperform smaller ones due to their greater capacity and computational resources. However, the computational dependency can be mitigated in practice through techniques such as distillation and quantization to reduce model size, or by leveraging cloud services to provide scalable compute resources for inference. These complementary methods effectively address computational limitations, ensuring that our framework remains promising and scalable for real-world applications.
>
> ### Q3: Comparison with Human Experts
>
> A: We appreciate this suggestion and conduct additional experiments for comparison with human experts and human-agent interaction.
>
> **Tab. R2: Human expert and interaction evaluation**
> | Task Name     | VIKI-L1↑  | VIKI-L2↑  | VIKI-L3 (RMSE) ↓ | VIKI-L3 (HD) ↓ | VIKI-L3 (DFD)  ↓|
> |---------------|----------|----------|-----------------|---------------|----------------|
> | Success Rate  | 98%      | 96.5%    | -               | -             | -              |
> | Metrics       | -        | -        |  48.84   | 58.97   | 71.3   |
>
> (1) Human Expert Evaluation: We involved 200 tasks from the test set, with human experts completing them. Tab.R2 shows that VIKI-R achieved high success rates in VIKI-L1 and VIKI-L2  (98% and 96.5%, respectively), and showed notable reductions in VIKI-L3 in RMSE, HD, and DFD compared to human performance.
>
> (2) Human-Agent Interaction: We involved 30 tasks from VIKI-L2, where humans first provided action sequences, and the model inferred the robot’s actions in the scene. The success rate was 36.67%, comparatively higher than current proprietary models (11.82%-17.50%). This demonstrates the potential of our approach in human-agent collaboration and highlights the opportunities for further refinement and optimization to achieve superior performance.
>
> ### Q4: Degradation of Reinforcement Learning without Chain-of-Thoughts Grounding
>
> A: Indeed, recent research has underscored the importance of Chain-of-Thoughts (CoT) grounding in enhancing the performance of Large Language Models (LLMs) in reinforcement learning, particularly in improving reasoning and decision-making capabilities. Our approach builds on this strategy by incorporating CoT grounding to achieve superior performance across the tasks in our benchmark. Although performance may degrade without this step, the grounding data necessary for this can be easily obtained using current multimodal models (e.g., GPT-4o) at a relatively low cost, which makes our method practical. While the CoT grounding paradigm could be applied to other supervised datasets in reinforcement learning, exploring such extensions falls within the broader field of reinforcement learning, and is beyond the scope of our work's main contribution.

---

> > ### Comment · Reviewer_UsEP · 2025-08-06
> >
> > The real-world deployment results (Tab. R1), along with new evaluations involving human experts and human-agent interaction (Tab. R2), address my initial concerns. These new experiments demonstrate the practicality of the benchmark, but also highlight key gaps between current model performance and human capabilities.

---

### Note · Authors · 2025-08-16

We sincerely thank the reviewers and the AC for their thoughtful feedback. We are encouraged that all reviewers recognized the novelty and impact of VIKI-Bench, the first hierarchical benchmark for embodied multi-agent cooperation, and our VIKI-R framework. We also appreciate that, after rebuttal and additional experiments, reviewers confirmed their concerns were addressed and indicated score increases.

### Key Contributions

Our work introduces (1) a hierarchical benchmark spanning agent activation, task planning, and trajectory perception; (2) a diverse dataset with heterogeneous robots, multi-view observations, and structured annotations; and (3) the VIKI-R framework, which combines CoT-grounded supervised fine-tuning and reinforcement learning with multi-level reward signals. These contributions fill a clear gap in evaluating multi-agent embodied intelligence and provide a strong foundation for future research.

### Addressed Concerns

- Real-world deployment: We presented preliminary physical experiments with real robots (dual-arm RealMan, Panda, Agilex) on VIKI-L1, demonstrating feasibility and highlighting the sim-to-real gap. We are extending this to VIKI-L2/L3.

- Human comparison: Expert and human-agent interaction studies show both potential and the gap between models and humans.

- Task difficulty and ablations: We analyzed difficulty sources (agent number, reasoning horizon), reported success rates, and added ablations on reward design, model scale, and optimization.

- Resource considerations: We detailed training costs and GPU usage, and clarified that GPT-4o is only used for data generation (~$0.05 per task).

- Benchmark utility: We provided comparisons with MAP-THOR/LLaMAR, ReAct, and MAD, along with visualizations and customization options. Tutorials and demos will be released in the official repository.

### Broader Impact

We acknowledge limitations such as simulation reliance and compute demands of large VLMs. Nonetheless, VIKI-Bench establishes a systematic framework for embodied multi-agent reasoning, while VIKI-R shows a practical approach that bridges supervised and reinforcement learning. The positive reviewer feedback and raised scores validate the novelty and value of this contribution.

We again thank the reviewers and AC for their constructive input. We are committed to integrating improvements and supporting the community through the public release of datasets, code, tutorials, and real-world extensions.

---

### Decision · Program_Chairs · 2025-09-18

**Decision:**

Accept (poster)

**Comment:**

### Summary

The paper presents a three-level hierarchical benchmark for embodied multi-agent cooperation (activation, task planning, trajectory perception) together with VIKI-R, a two-stage fine-tuning framework that first warm-starts with supervised CoT-style traces and then applies RL with multi-level rewards. Experiments show consistent gains over SFT-only and RL-only baselines and competitive comparisons to strong open/closed models. The dataset and code are released to support community use, although more detailed tutorials will be added later to the public repo.

### Strengths
- **Hierarchical benchmark structure.** Reviewers highlighted that the benchmark's structured levels providing a clear decomposition of multi-agent cooperation (UsEP, mtDG, dVhv).
- **Method improves over baselines.** Reviewers reported that the proposed VIKI-R pipeline improves over supervised-only and RL-only baselines across the sub-tasks, with comparative results viewed as thorough (reported by UsEP; "clear improvements over baselines" noted by Ktr9; "comprehensive experimental setup" noted by dVhv).
- **Relevance and scope.** The focus on embodied, multi-agent settings with heterogeneous embodiments and both global/egocentric views addresses an area with limited standardized evaluation (UsEP, Ktr9).
- **Release for community use.** Reviewers acknowledged that data and code are publicly released, enabling replication, even as they asked for stronger tutorials (availability appreciated by dVhv; documentation requests by mtDG).

### Weaknesses
- **Early real-world breadth.** Real-world validation is primarily at L1; evaluation of L2/L3 on physical robots remains to be shown (raised by UsEP).
- **Data-generation sources/cost.** Some training traces rely on an external LLM (e.g., GPT-4o) for planning, prompting questions about costs, stability, and reproducibility if APIs change (raised by dVhv).
- **Usability/documentation.** Reviewers requested end-to-end tutorials, demos, and broader ablations/visualizations to ease adoption; current documentation was considered **thin** relative to prospective users' needs (raised by mtDG).

### Discussion & Rebuttal Phase
- UsEP: Asked for real-world validation and human comparisons; after the authors reported preliminary hardware results for the activation level and human-expert / human–agent studies, UsEP said these addressed their core concerns and confirmed accept.
- Ktr9: Requested analyses on difficulty sources, success rates, and relations among the three benchmark levels; after the authors' additions, Ktr9 raised their score and marked issues resolved.
- dVhv: Sought clearer positioning/ablations and clarity around the origin/cost/stability of LLM-generated planning traces; after detailed responses, dVhv increased their score.
- mtDG: Requested broader baselines/ablations/visualizations and end-to-end tutorials/demos; following author commitments and added analyses, mtDG raised to borderline-accept.
- Authors' final remarks: Reiterated public release (data/code), summarized new evaluations (hardware L1; human studies), and committed to improved tutorials/demos.

### Justification of final recommendation
The hierarchical benchmark and the proposed method VIKI-R that improves over supervised-only and RL-only baselines across sub-tasks, aligns with the Datasets & Benchmarks goal of enabling and accelerating ML research. The rebuttal meaningfully strengthened the case (preliminary real-world results, human studies), addressing key reviewer concerns. Remaining limitations--early breadth of real-world validation, and clearer guidance on the origin/cost/reproducibility of LLM-generated traces--are acknowledged and actionable.

However, the documentation available doesn't facilitate usability by the community yet. The authors reference Sec. C.2 and D in the supplementary for details on experimental setup, but I find these to be too brief to aid user adoption. In the Datasets & Benchmarks track, the bar centers on artifacts whose code/data are *accessible, documented, and executable* (as described in the CFP). While the data and repo are certainly available, stronger tutorials would make the resource more accessible to the community, and prevents me from recommending a higher Accept.